# Wide-field mid-infrared hyperspectral imaging beyond video rate

Jianan Fang[1], Kun Huang [1,2,3] ✉, Ruiyang Qin[1], Yan Liang[4], E Wu[1,2], Ming Yan [1,2] & Heping Zeng [1,2,5,6] ✉

Mid-infrared hyperspectral imaging has become an indispensable tool to spatially resolve chemical information in a wide variety of samples. However, acquiring three-dimensional data cubes is typically time-consuming due to the limited speed of raster scanning or wavelength tuning, which impedes real-time visualization with high spatial definition across broad spectral bands. Here, we devise and implement a high-speed, wide-field mid-infrared hyperspectral imaging system relying on broadband parametric upconversion of high-brightness supercontinuum illumination at the Fourier plane. The upconverted replica is spectrally decomposed by a rapid acousto-optic tunable filter, which records high-definition monochromatic images at a frame rate of 10 kHz based on a megapixel silicon camera. Consequently, the hyperspectral imager allows us to acquire 100 spectral bands over 2600-4085 $cm^{-1}$ in 10 ms, corresponding to a refreshing rate of 100 Hz. Moreover, the angular dependence of phase matching in the image upconversion is leveraged to realize snapshot operation with spatial multiplexing for multiple spectral channels, which may further boost the spectral imaging rate. The high acquisition rate, wide-field operation, and broadband spectral coverage could open new possibilities for high-throughput characterization of transient processes in material and life sciences.

Hyperspectral imaging is a prominent non-invasive analytical technique that allows for the simultaneous acquisition of both spatial and spectral information[1,2]. Particularly, mid-infrared (MIR) hyperspectral imaging has attracted increasing attention due to the richness of material and molecular signatures observable in this spectral range[3]. Vibrational imaging methods for spectral mapping provide a complementary and more sensitive tool when compared with Raman spectroscopic counterparts[4,5] because infrared absorption offers a much larger cross-section than that of Raman scattering[6]. Nowadays, MIR spectral imaging technologies have been widely used in chemical, medical, and bio-related fields, such as non-destructive material

detection, stand-off gas analysis, high-throughout environmental monitoring, and label-free biomedical diagnosis[7–10]. Despite tremendous successes in these applications, MIR hyperspectral imaging has long been plagued by the time-consuming acquisition of three-dimensional spectral data cubes[11], which becomes more pronounced for demanding requirements with high spatial definition and broad spectral bands. The available speed for existing hyperspectral imaging instruments is prohibitively slow for rapid analysis or in situ observations of transient processes, which is pertinent to time-sensitive scenarios including fast characterization of gaseous combustion, high-throughput sorting of biomedical samples, and real-time tracking of

[1]State Key Laboratory of Precision Spectroscopy, East China Normal University, Shanghai 200062, China. [2]Chongqing Key Laboratory of Precision Optics, Chongqing Institute of East China Normal University, Chongqing 401121, China. [3]Collaborative Innovation Center of Extreme Optics, Shanxi University, Taiyuan, Shanxi 030006, China. [4]School of Optical Electrical and Computer Engineering, University of Shanghai for Science and Technology, Shanghai 200093, China. [5]Shanghai Research Center for Quantum Sciences, Shanghai 201315, China. [6]Chongqing Institute for Brain and Intelligence, Guangyang Bay Laboratory, Chongqing 400064, China. ✉e-mail: khuang@lps.ecnu.edu.cn; hpzeng@phy.ecnu.edu.cn

living organisms[8,12–14]. In this context, there is an urgent need to realize high-speed MIR spectral imaging associated with dense sampling points in both spatial and spectral domains.

As a benchmarking technique for chemical imaging, Fourier transform infrared (FTIR) imaging spectrometers have witnessed tremendous progress over the past decades[15]. To boost the data acquisition speed, FTIR spectral imagers mostly operate in a wide-field fashion based on focal plane arrays (FPAs)[13,16]. However, the limited illumination intensity of commonly used globar thermal sources may necessitate a long dwell time for detection or an averaging operation for denoising to obtain a better signal-to-noise ratio (SNR)[15]. These procedures inevitably restrict the acquisition rate, especially for large format detectors, due to a low divided light flux among each pixel[17]. Although measurements with an improved SNR can resort to higher-brightness broadband sources based on synchrotron radiation[18] or laser supercontinuum[19], the refreshing speed for the FTIR spectrometer intrinsically suffers from the mechanical scanning required to record interferograms, which leads to a total acquisition time of up to minutes[13].

Recently, an alternative approach based on quantum cascade lasers (QCLs) has emerged to implement MIR spectral imaging at an improved speed due to the agile spectral tuning capability for the light source[8,10,17]. For instance, the frame rate of QCL-based imaging at a single wavenumber reaches 50 Hz in 640 × 480 pixels[8], which permits real-time visualization of datacubes only for several spectral bands. Rapid access of more spectral points is typically comprised of a reduced number of active pixels, thus resulting in a smaller field of view or a degraded spatial resolution[17]. Indeed, MIR detectors based either on microbolometers or narrow-bandgap semiconductors have been plagued with high dark noise, low pixel count, and limited frame rate[20,21]. Such a bottleneck in wide-field acquisition rate is also manifested in the scheme based on a MIR supercontinuum source and an acousto-optic tunable filter (AOTF)[22,23]. Specifically, the collection of an image cube of 100 wavelengths and 300 k pixels takes as long as 2 s[22]. To date, it is still a long-sought-after goal to realize a video-rate MIR hyperspectral imaging at high definition, i.e., reaching a datacube refreshing rate over 24 Hz for a spatial format in a megapixel scale, which calls for faster acquisition and image processing speed.

To this end, the so-called upconversion imaging approach has been developed to circumvent the limitations of current MIR direct imagers[24–29]. In this approach, the MIR information is typically transferred to the visible or near-infrared replica, where a sensitive and fast detection can be realized by leveraging high-performance large-bandgap sensors[24]. Such upconversion configuration has led to superior room-temperature MIR imaging performances with single-photon sensitivity[25,30] and sub-megahertz frame rate[31,32] based on high-definition silicon cameras. So far, there have been various schemes proposed to implement the MIR hyperspectral upconversion imaging[24]. One straightforward strategy is to employ tunable light sources based on an optical parametric oscillator (OPO)[33]. In this fashion, a video-rate MIR monochromatic operation has been demonstrated in a wide-field mode[33], yet the achievable spectral imaging speed is limited by the wavelength tuning rate over a wide-spanning spectrum. Notably, QCL sources have been adopted for MIR upconversion spectral imaging, but the reported frame rate is limited by a relatively long acquisition time of up to 10 ms due to the low conversion efficiency in the continuous-wave pumping scheme[34]. Alternatively, broadband illumination can be used to capture the full spectral information. However, a scanning procedure and subsequent post-processing are often needed to extract single-wavelength components from the polychromatic upconverted images, such as relying on temperature variation[25,35] or angular rotation[34] of the nonlinear crystal, as well as spatial translation of the object scene[36]. A promising remedy to achieve a faster spectral decomposition has recently been reported based on a visible-band hyperspectral camera, which allows

to capture of monochromatic images with 640 × 480 pixels at a frame rate over 100 Hz[37]. The total acquisition time is about 8 s for recording a spectral data cube with dense wavelength channels, which is intrinsically limited by the scanning speed of the interior grating.

In parallel, other promising architectures for indirect MIR imaging have also been intensively investigated by resorting to the photothermal effect within interrogated samples[6,38,39] or two-photon absorption (TPA) of the optical detector itself[40,41]. For instance, a wide-field photothermal microscope enables a chemical imaging speed at video rate[39], albeit the spectral sweeping rate is hampered by the wavelength tuning time of the involved OPO source. Additionally, TPA-based spectral imaging at high definition is implemented by combining time-stretched spectroscopy and delay-scanning optical gating, which facilitates a 1.1-s acquisition time per spectral sweep over 530 cm$^{-1}$ [41]. In previous reports, the ultimate spectral imaging speed is impeded by deficiencies in rapid spectral filtering and/or fast wide-field detection, which is insufficient to observe transient phenomena and real-time processes.

Here, we propose and demonstrate a wide-field MIR hyperspectral upconversion imaging at high speed and high definition over a spectral range of 2600–4160 cm$^{-1}$. The upconversion imaging features a wide-field spatial mapping and a high-fidelity spectral conversion, thus preserving the full infrared absorption information in the spatial and spectral dimensions. The resulting polychromatic upconverted field is rapidly filtered by an AOTF in the visible region, which permits to recording of high-definition monochromatic images at frame rates over 10 kHz based on a megapixel silicon camera. Consequently, 100 channels of spectral images can be collected in 10 ms, corresponding to a refreshing rate of up to 100 Hz. The achieved spectral imaging speed is over two orders of magnitude faster than previously reported results at comparable spectral channels and spatial formats. Furthermore, spatial multiplexing in the upconverted image is investigated based on the angular dependence of phase matching in the nonlinear upconversion, which offers an approach to augment the imaging speed. Therefore, our work addresses the long-standing quest to implement video-rate acquisition of MIR hyperspectral data cubes at high-pixel density, which would stimulate immediate applications in high-throughput characterization of dynamic processes in material and life sciences.

## Results
### Basic principle

The operation procedures of the proposed MIR hyperspectral imaging are conceptually illustrated in Fig. 1A, which consists of broadband nonlinear conversion, rapid spectral filtering, and wide-field detection. Specifically, the conversion step constitutes an essential core to bridge the wavelength discrepancy between the infrared radiation with label-free chemical specificity and the visible band compatible with high-performance silicon detectors[24]. The involved conversion process is required to preserve the spatial and spectral information in a snapshot fashion, which thus calls for a wide-field and broadband frequency upconverter. To this end, a chirped-poling lithium niobate (CPLN) nonlinear crystal is devised in our upconversion architecture, which permits to satisfaction of the phase-matching condition for incident signals over a large acceptance angle and a broad spectral range[28,31]. The resulting wide-field operation is critical to implement a fast snapshot acquisition, which contrasts with previous scanning-assisted modalities based on temperature tuning[25,35], crystal rotation[33,34], or object translation[36]. The wide-field supremacy has been manifested in high-speed MIR imaging within a narrow spectral band[31,42], but the potential of simultaneous broadband capability has not yet been revealed in the realm of spectral imaging. In parallel, high-fidelity spectral mapping is required to preserve the infrared absorption profile at each spatial coordinate. The pump bandwidth will result in a resolution degradation due to the spectral convolution deformation. It

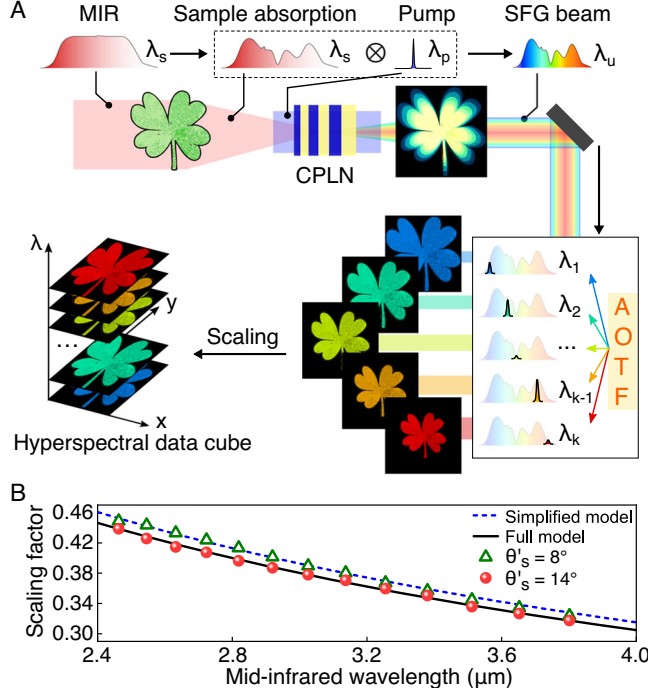

**Fig. 1 | Conceptual illustration of the wide-field hyperspectral upconversion imaging. A** A broadband mid-infrared (MIR) source is used to illuminate a sample. The generated absorption pattern then passes through a wide-field and broadband upconverter based on a chirped-poling lithium niobate (CPLN) nonlinear crystal. A narrow-band pump ensures a high-fidelity mapping of the spectral information through sum-frequency generation (SFG). The resultant upconversion replica is radially blurred due to wavelength-dependent spatial scaling factors governed by the phase-matching condition. Consequently, an acousto-optic tunable filter (AOTF) is employed to retrieve monochromatic images with a fast scanning speed and a high spectral resolution. Note that the inertia-free tuning ability for the AOTF allows one to address arbitrary single or multiple spectral channels in an agile fashion. Finally, a spectral data cube can be obtained after a simple rescaling operation. **B** Spatial scaling factor for the upconverted spectral image as a function of the signal wavelength for the experimental measurement and theoretical calculation. Note that the scaling factor generally depends on the incident angle. The presented measurements and simulations are conducted at an incident angle at 8° and 14°. Dashed and solid lines correspond to small-angle approximation and rigorous models, respectively.

is thus imperative to devise a narrow-band pump for improving the spectral correspondence[32].

Consequently, the full information in spatial and spectral domains is completely transferred into the upconversion replica in the visible region. In our approach, an AOTF is used to quickly extract a monochromatic image at an arbitrary wavelength, which eliminates the need for any moving part or data post-processing as required in previous demonstrations[34–36]. Indeed, the AOTF serves as an electronically-controlled bandpass optical filter, which features large optical throughput, agile spectral tuning, and a programable random-access ability[23]. In general, the required driving power for a maximum diffraction efficiency quadratically scales with the wavelength[43]. As a result, a large-aperture AOTF at MIR wavelengths typically operates with a compromised efficiency by taking into account the available radio-frequency power and achievable switching speed[43]. Notably, AOTFs beyond 4.5 $\mu$m are not commercially accessible. Here, the synergic combination of an infrared upconverter and a visible-band AOTF provides a way to implement high-speed MIR spectral imaging based on a high-definition silicon camera.

In the proposed scheme, the frequency upconversion is implemented at the Fourier plane of the targeted object, which gives rise to a

blurring effect in the polychromatic upconverted image due to wavelength-dependence spatial magnifications for various spectral components[24,28]. The relevant mechanism lies in the momentum conservation in the transverse direction during the sum-frequency generation (SFG) process, i.e., nulling the phase mismatch $\Delta k_\perp = k_u \sin \theta_u - k_s \sin \theta_s$, where $k_{s,u}$ are wave vectors for the signal and upconverted light within the crystal, and $\theta_{s,u}$ denote the corresponding beam angles relative the propagation axis. The angular magnification between the external angles $\theta'_{s,u}$ is simply given by

$$\frac{\sin \theta'_u}{\sin \theta'_s} = \frac{\lambda_u}{\lambda_s}, \tag{1}$$

where Snell's law is used at the air-medium interface. For a 4-f imaging system formed by two relay lens with focus lengths of $f_1$ and $f_2$, the imaging scaling factor between the image and object planes in the small-angle approximation is deduced to be

$$M(\lambda_s) \approx \frac{f_2 \sin \theta'_u}{f_1 \sin \theta'_s} = \frac{f_2 \lambda_u}{f_1 \lambda_s} = \frac{f_2 \lambda_p}{f_1 (\lambda_s + \lambda_p)}, \tag{2}$$

where the energy conservation law $\lambda_u^{-1} = \lambda_s^{-1} + \lambda_p^{-1}$ is used. Figure 1B presents the measured spatial magnification as a function of the MIR signal wavelength, which agrees well with the theoretical line given by the simplified model for incident angles up to $\theta'_s = 8°$. In the presence of larger angles, a more rigorous model, as detailed in Supplementary Note 1 should be used to ensure a proper scaling factor, as verified by the experimental observation at $\theta'_s = 14°$. In practice, an accurate scaling map can be pre-determined to facilitate a fast spatial normalization for each monochromatic image at a known filtering wavelength.

## Imaging setup and characterization

Figure 2 presents the experimental setup for the MIR hyperspectral imaging based on broadband nonlinear frequency conversion. The involved MIR source originates from a high-brightness supercontinuum fiber laser with spectral coverage from 1.9 to 3.9 $\mu$m[19]. The supercontinuum source eliminates the need for tuning operation as typically required for OPO sources[37,41] in broadband imaging. A long-pass filter is used to select the wavelength portion longer than 2.4 $\mu$m, as shown in Fig. 3A. The repetition rate of MIR pulses is about 2.35 MHz. The high operation frequency favors subsequent high-speed imaging at a high frame rate and a short exposure time. Meanwhile, a pump source is prepared from a gain-switched laser diode at 1064 nm. The semiconductor laser is synchronously gated by an electrical pulse generator with a timing trigger from the MIR pulse. The pulse duration and time delay can be electronically controlled at a precision of 1 ps, thus providing a simple and robust alternation to dual-color sources based on OPOs[28,33] or mode-locked lasers[30,44]. The pump power is boosted to about 600 mW, and the spectral width of the amplified pulse is kept below 0.2 nm with the help of a large-mode-area fiber amplifier. The spectro-temporal engineering of the involved synchronous pulses is essential to facilitate high-fidelity spectral mapping in a high-efficiency and low-noise fashion[32,44]. Detailed description of the laser sources for the coincidence-pumping nonlinear upconversion is presented in Methods.

The MIR beam is expanded to illuminate the sample before being steered into a 4f imaging system. A CPLN nonlinear crystal is placed at the Fourier plane to perform the SFG process under a spatially and temporally overlapped pump pulse. The CPLN is fabricated with a linearly-ramping polling period from 16 to 24 $\mu$m, which permits a wide-field and broad-spectrum upconversion imaging due to the self-adapted quasi-phase-matching for various incident angles and signal wavelengths[28,31,42]. Hence, the spatial and spectral information is simultaneously transferred to the polychromatic upconverted field in

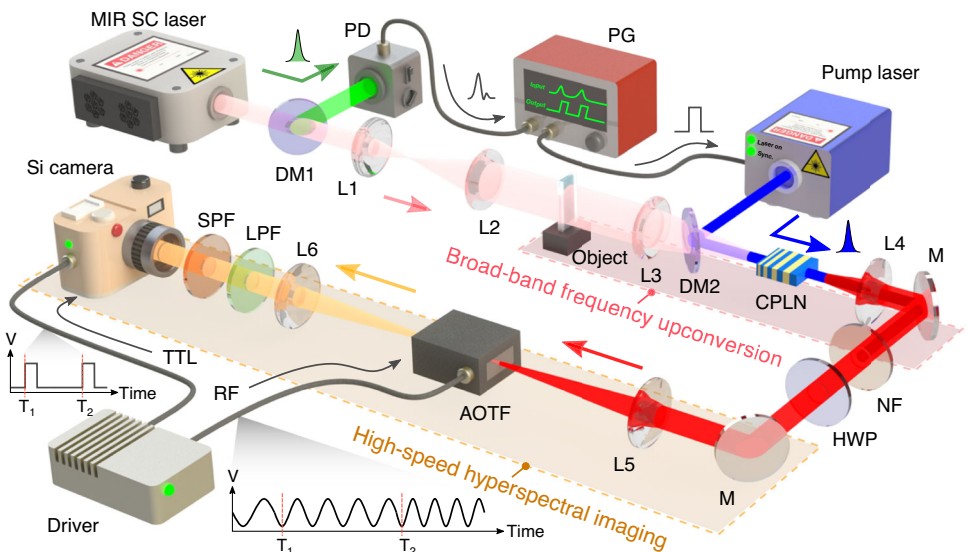

**Fig. 2 | Experimental setup.** A MIR supercontinuum (SC) pulsed laser is used to provide high-brightness illumination over broadband spectral coverage. The infrared beam passes through a dichroic mirror (DM) that allows a high transmission for wavelengths above 2.4 μm. The reflected portion is detected by a photodiode (PD), which serves as a trigger for generating a synchronous pump pulse at 1064 nm. The optical pulse duration and temporal delay can be controlled by an electrical pulse generator (PG). The infrared field after a sample is steered into a 4f imaging system, where a chirped poling lithium niobate (CPLN) crystal is placed at the Fourier plane to perform a sum-frequency generation based on the coincident pumping. The upconverted field is then sent through an acousto-optic tunable filter (AOTF) before being captured by a silicon camera. The involved timing sequences for the spectral tuning and imaging recording are managed by a programmed driver that delivers TTL and radio-frequency (RF) signals. L: lens; M: silver mirror; SPF, LPF, and NF: short-, long-pass, and notch filters.

the visible region, as illustrated in Fig. 3B. The upconverted image is then relayed into a large-aperture AOTF with a diffraction efficiency over 85%, a spectral resolution of about 2.9 nm, a switching time of 2.8 μs (see Methods). The filtered image is captured by a high-definition silicon camera with a maximum frame rate of over 200 kHz[31], which allows us to implement ultra-high-speed monochromatic imaging. Finally, real-time spectral imaging beyond video rate can be realized without the necessity for complex image processing or reconstruction. The electronic timing sequences and radio-frequency (RF) generation are assisted by a controlling driver based on a field programmable gate array (FPGA). More discussions on the imaging setup and operation configuration can be found in Supplementary Notes 2 and 3.

The imaging performance is characterized by using a USAF-1951 resolution target. Figure 3C presents the recorded image without the AOTF, which exhibits a radial blurring effect due to wavelength-dependent spatial scaling factors as predicted by Eq. (2). Typically, a deblurring procedure is required to obtain monochromatic images, which has been realized in previous demonstrations based on parameter variation of the phase-matching condition[34,35] or mechanical scanning of a dispersive grating[37]. However, these tuning methods are time-consuming, which hence severely limits the spectral imaging speed. Instead, the AOTF adopted here is inertial-free, and permits an agile operation within several μs. Representative monochromatic images are displayed in Fig. 3D–F for three disparate MIR wavelengths at 3508, 2980, and 2595 nm. The resultant wavelength-dependence spatial magnification can be numerically compensated with the pre-determined scaling factor, as explained in Supplementary Note 5.

Accordingly, the filtered spectra in Fig. 2G–I indicate a full width at half maximum (FWHM) of about 2.9 nm. As discussed in Supplementary Note 3, the AOTF resolution can be enhanced to 0.9 nm by using a longer acousto-optic crystal, albeit with a longer switching time of about 8 μs. Correspondingly, the MIR spectral resolution is improved to about 15 cm⁻¹, which is sufficient to characterize solid and liquid samples. Moreover, the AOTF has a unique feature of parallel filtering,

which allows us to simultaneously select multi-channel spectral components as shown in Fig. 3J–O. The abilities of random hopping and wavelength multiplexing could be used to facilitate flexible filtering functionalities in spectral imaging acquisition.

## Wide-field MIR upconversion spectral imaging

Now we turn to characterize the MIR hyperspectral imaging performance by examining static scenes with chemical and morphological information. As illustrated in Fig. 4A, two types of plastic films are prepared to cover a copper sheet that is engraved with the acronym of "ECNU". Film1 and film 2 are taken from two kinds of adhesive tapes, which are made of biaxially oriented polypropylene and cellulose acetate, respectively. The AOTF is set to scan 105 spectral bands with a wavenumber step of 15 cm⁻¹ over the spectral range of 2600–4160 cm⁻¹. The measured spectra for the two polymer films are presented in Fig. 4B, which agree well with the FTIR references. Particularly, three spectral bands at 2915, 3500, and 3995 cm⁻¹ are selected to display the corresponding monochromatic images, as shown in Fig. 4C. The two films can simply be differentiated according to different absorptions at 3500 cm⁻¹.

Besides of material identification, spectral imaging is commonly used to quantify the chemical concentration. In this case, another liquid sample is prepared by gently injecting a drop of ethyl alcohol ($C_2H_5OH$) into the carbon tetrachloride ($CCl_4$) solvent within a cuvette. Figure 4D presents the monochromatic image at 3600 cm⁻¹, where a gradient transition of the concentration is manifested along the depth axis. The full spectral imaging cube is illustrated in Fig. 4E, which indicates two absorption peaks at 2900 and 3320 cm⁻¹. Figure 4F shows the absorbance profiles for four representative points at various depths. The measured absorbance at 3320 cm⁻¹ can be used to quantitatively identify the vertical concentration distribution, as shown in Fig. 4G.

Notably, the inertial-free AOTF allows for a non-uniform sampling strategy, where the point number and wavelength interval can be adapted to focus specific spectral regions of interest. Such a foveated operation is especially attractive in object classification and

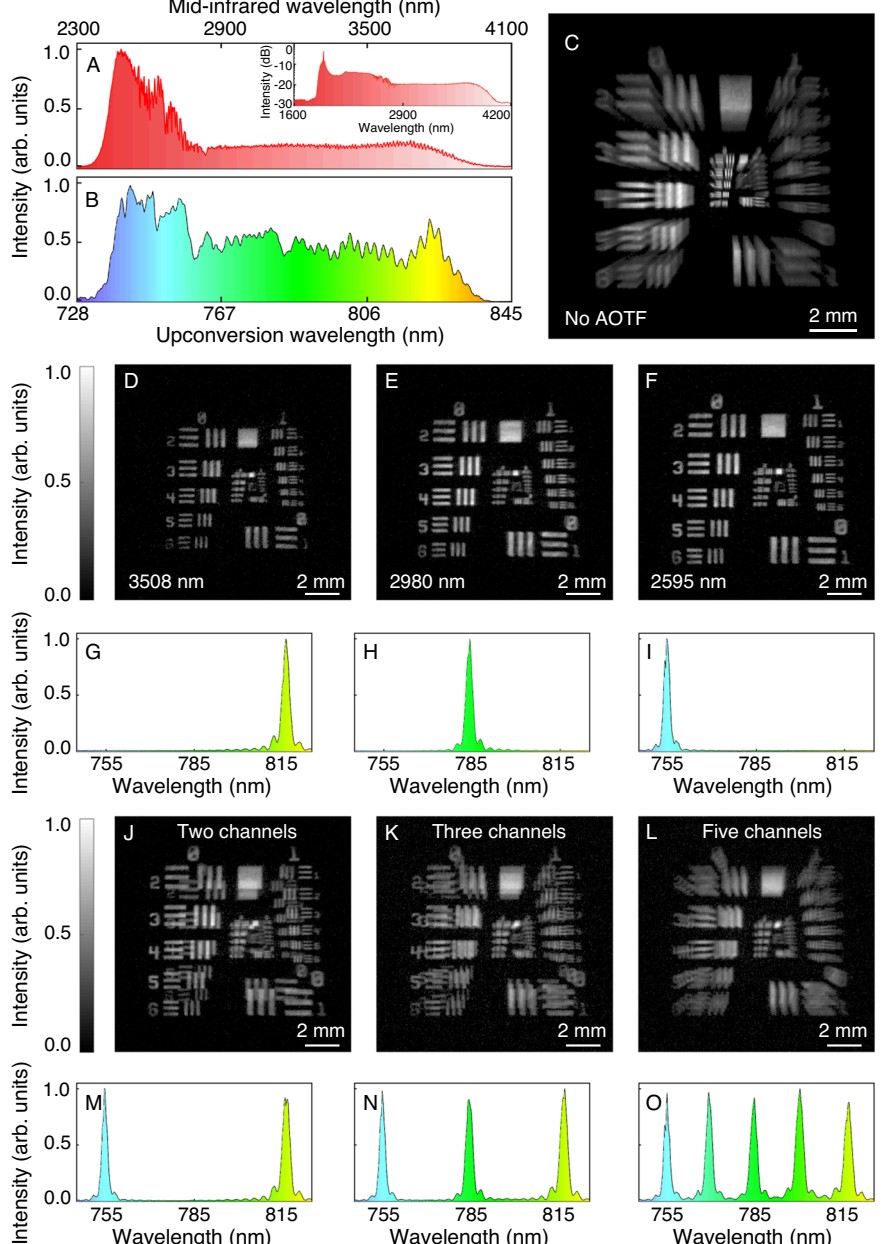

**Fig. 3 | MIR upconversion spectral imaging based on the acousto-optic filtering. A** Optical spectrum of the broadband MIR light after a long-pass filter with a cut-off wavelength at 2.4 μm. **B** Optical spectrum of the frequency-upconverted field. **C** Recorded image for a resolution test target without the AOTF. The exhibited radial blurring effect is ascribed to the overlap of all spectral components with various spatial scaling factors. **D**–**F** Monochromatic images at 3508 nm (**D**), 2980 nm (**E**), and 2595 nm (**F**) with the AOTF. **G**–**I** Corresponding filtered spectra for the upconverted field at 816 nm (**G**), 785 nm (**H**), and 754 nm (**I**), respectively. **J**–**L** Spectral images with two (**J**), three (**K**), and five (**L**) channels based on the parallel filtering operation of the AOTF. **M**–**O** Corresponding filtered spectra for the upconverted field.

concentration evaluation by leveraging distinct spectral features associated with certain constituents.

## High-speed MIR hyperspectral videography

Next, we investigate the high-speed performance of the wide-field MIR hyperspectral imaging system. To this end, a dynamic scene is emulated by injecting a stream of ethanol into the $CCl_4$ solvent through a syringe needle. A full set of spectral images is acquired by uniformly scanning the AOTF from 2600 to 4085 $cm^{-1}$ with a step of 15 $cm^{-1}$. The resulting 100 spectral channels are numerated by the frame number $n$. The dwell time at each channel is 100 μs, which is mainly determined by the exposure time of the camera. The total acquisition time is thus as little as 10 ms, corresponding to a refreshing rate of 100 Hz. The

AOTF is continuously operated in a cyclic fashion, which generates a sequence of spectral data cubes. Each data set is enumerated by the sequence number $m$. The elapsed time can be calculated by $T = (m \times 10 + n \times 0.1)$ ms.

Figure 5A presents the recorded sequence $m = 130$, which demonstrates transmission patterns for a series of spectral channels. The morphological distribution of the injected ethanol is almost kept unchanged within the short acquisition time of 10 ms. The fast MIR spectral videography is crucial to extracting the spectral information in a dynamic scene. It can be seen that the frames at $n = 20$ and 48 exhibit relatively dark patterns, where the illumination wavelengths are close to the two absorption peaks at 2900 and 3320 $cm^{-1}$, as shown in Fig. 4F. Intriguingly, the chemical

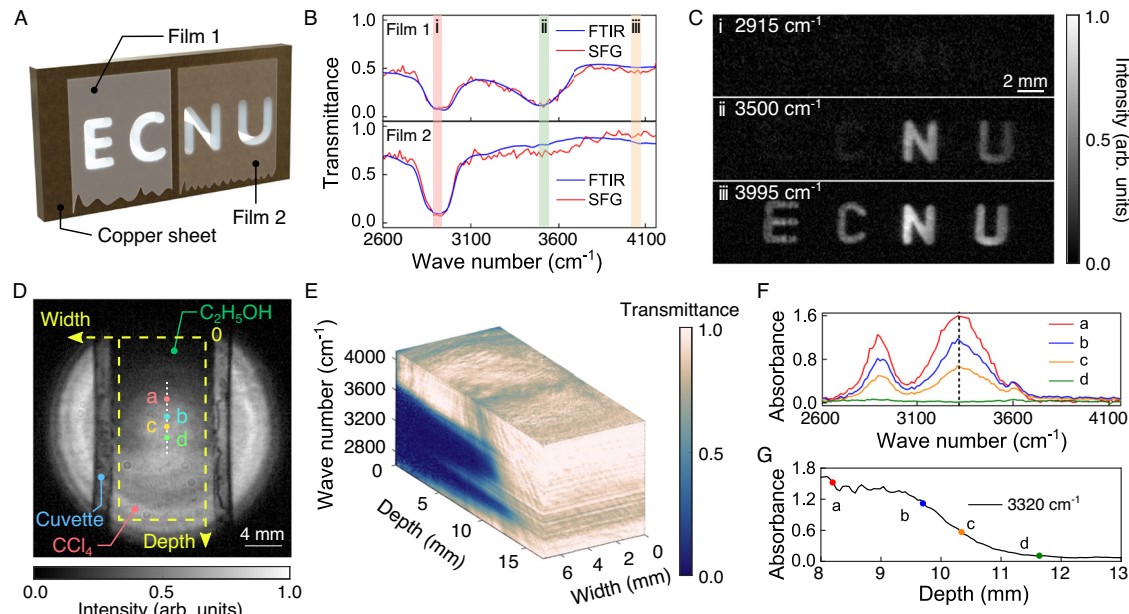

**Fig. 4 | Wide-field MIR hyperspectral imaging for polymer and liquid samples.** **A** Sample preparation with two types of plastic films that cover four letters carved on a piece of sheet copper. **B** Measured infrared absorption spectra for the two films, which agree well with the ones given by the FTIR. **C** Monochromic images at three selective spectral bands at 2915, 3500, and 3995 cm⁻¹, which can be used for material identification based on distinct absorption profiles of the samples. **D** Monochromatic image at 3600 cm⁻¹ for a mixed liquid where a drop of ethyl alcohol ($C_2H_5OH$) is injected into the carbon tetrachloride ($CCl_4$) solvent within a cuvette. **E** Measured hyperspectral image for the vertical section of the mixed liquid. **F** Measured spectral absorbances at various depths indicated by the positions in (**D**). **G** Measured absorbance at 3320 cm⁻¹ as a function of the depth, which indicates a gradual transition for the ethanol concentration.

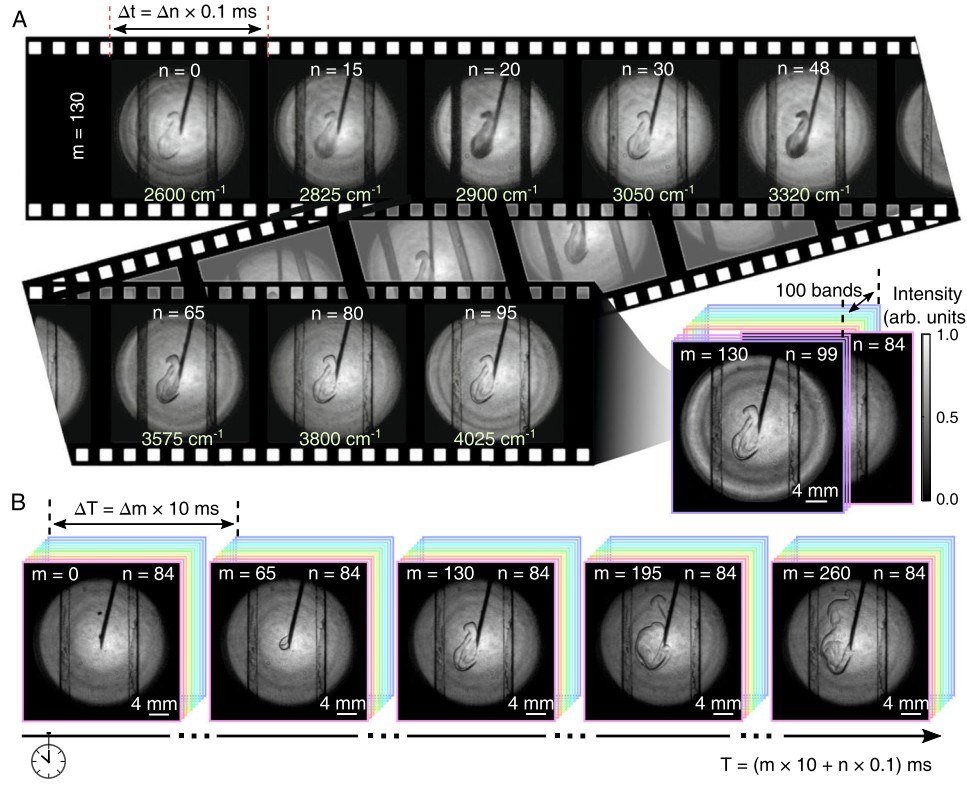

**Fig. 5 | High-speed MIR spectral videography of liquid injection processes. A** A stream of ethyl alcohol ($C_2H_5OH$) is injected into the carbon tetrachloride ($CCl_4$) solvent through a syringe needle. The hyperspectral imaging is measured at a wavenumber step of about 15 cm⁻¹ for 100 bands from 2600 to 4085 cm⁻¹. The exposure time at each spectral band is set to 0.1 ms to produce high-contrast monochromatic images. The full set of spectral images is given in Supplementary Movie 1. **B** The spatial evolution of injected liquid is examined at the wavelength channel at 3860 cm⁻¹, corresponding to the frame $n = 84$ within each set of spectral data cubes. A 100-Hz refreshing rate for the specific band is determined by the circling time of 10 ms, which enables us to capture the transient diffusion process in real-time.

absorption of the cuvette edge is also manifested in the sequence of monochromatic images, which indicates similar absorption bands for hydrocarbons. The corresponding full set of spectral images is recorded in Supplementary Movie 1. Furthermore, a real-time visualization of the liquid injection process is demonstrated in Fig. 5B, where a specific frame at $n = 84$ is taken from each sequence of data cubes. At the chosen wavenumber at 3860 cm$^{-1}$, the ethanol is almost transparent. The darkened contour is ascribed to the cast shadows in the non-uniform medium. The MIR shadowgraph is useful to reveal temporal and spatial information in flow dynamics at a chosen spectral band.

Finally, we proceed to demonstrate real-time hyperspectral imaging for liquid mixing dynamics. In this scenario, ethanol and benzene are simultaneously injected into the CCl$_4$ solvent through two syringe needles. Figure 6A gives the measured transmission spectra for the two types of liquid samples, which agree well with the references given by the NIST database. The ethanol and benzene have distinct absorption peaks at 3050 and 3350 cm$^{-1}$, which correspond to the frames $n = 30$ and 50 in the hyperspectral dataset, respectively. Figure 6B shows a shadowgraph for the mixed liquid at the spectral channel of 3800 cm$^{-1}$ for the frame $n = 80$. To better visualize the chemical differentiation, the dimensionality of the hyperspectral data stack can be reduced to two featured frames, where one sample appears transparent while another exhibits strong absorption and vice versa[41]. As illustrated in Fig. 6C, D, these two frames are represented as the red ($R = 255$, $G = 0$, and $B = 0$) and green ($R = 0$, $G = 255$, and $B = 0$) channels, which can be merged to produce a single image according to the RGB color format. Figure 6E presents the real-time sequences for the mixing dynamics, where the chemical contrast is manifested by the overlay of the red and green frames. The recorded video is given in Supplementary Movie 2.

## Discussion

MIR hyperspectral imaging has long been recognized as a central tool in the non-invasive identification of chemical constituents, along with the capability for simultaneous visualization of morphological structures. However, the operation speed is severely limited, especially for acquiring spectral data cubes with large spatial formats and broad spectral channels, which impedes high-speed analysis or real-time observations of dynamic samples. To date, it remains challenging to realize video-rate MIR spectral imaging with high definition and broad channels. In this work, we have addressed the long-standing quest to implement a wide-field MIR hyperspectral imaging beyond video rate. The spectral imaging speed is made possible by the collective innovations in bright supercontinuum illumination, broadband frequency upconversion, rapid acousto-optic filtering, and fast wide-field detection. The achieved performances here have significantly surpassed the reported benchmarks among various MIR spectral imaging techniques (see Supplementary Note 7).

In the spatial domain, the wide-field imaging modality significantly reduces the acquisition time of volumetric data cubes relative to the point-scanning fashion. The involved nonlinear upconverter allows us to obtain a wide field of view in one shot, which contrasts with previous works that need parameter scanning and post-processing[33–36]. The wide-field upconverted images are captured by using a fast and sensitive silicon camera with a megapixel frame matrix. Specifically, high-definition images can be recorded at a frame rate of up to 10 kHz, which is orders of magnitude faster than that for state-of-the-art infrared focal plane arrays[8,17]. In addition, the MIR supercontinuum laser is essential for recording high-SNR images within an exposure time down to tens of μs since it offers a much higher spectral radiance with a comparison to illumination sources based on thermal radiation[34–36] and parametric generation[26,27]. To go beyond the achieved spectral imaging

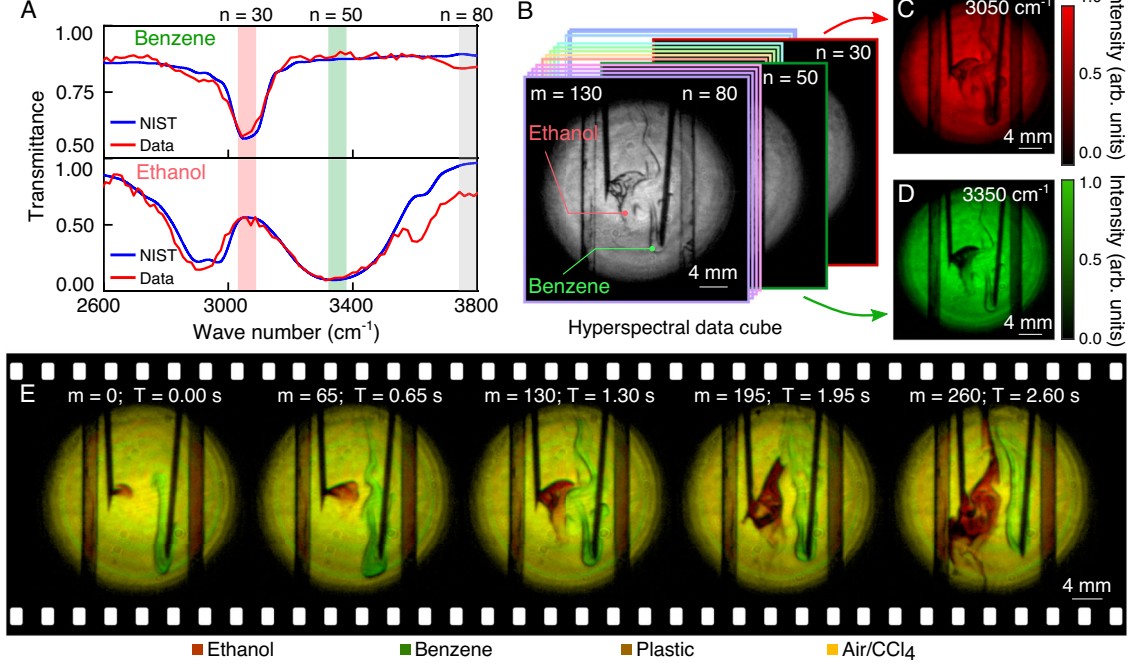

**Fig. 6 | Real-time MIR hyperspectral imaging of liquid mixing dynamics.**
**A** Transmission spectra of benzene and ethanol extracted from hyperspectral data cubes, which are consistent to the references given by the NIST database. Two specific bands at $n = 30$ and 50 are chosen to differentiate the two types of liquid, which correspond to the absorption peaks at 3050 and 3350 cm$^{-1}$ for the benzene and ethanol, respectively. **B** A-frame at $n = 80$ within the spectral data cube $m = 130$, which shows one representative snapshot during the liquid mixing process.

**C, D** Two frames at $n = 30$ and 50 are illustrated with colormaps in red (**C**) and green (**D**), respectively. **E** False-color visualization of spectral imaging with the integration of two distinct frames according to the RGB color format, which shows the mixing dynamics at a frame rate of 100 Hz. Note that each frame is recorded with 0.1 ms, while one set of data cubes is acquired in 10 ms for 100 spectral bands. A recorded video of the dynamic process is given in Supplementary Movie 2.

rate, a sufficient radiation flux should be acquired within a shorter frame time, which requires augmenting the illumination power or increasing the conversion efficiency. Currently, the conversion efficiency is about 0.01%, which can be improved by increasing the pump peak power or using a longer nonlinear crystal.

Another key advance in this work lies in the significant boost for the filtering speed in the spectral domain. The involved filtration is implemented based on an electronically-controlled AOTF, which features a large optical throughput, agile spectral tuning, and a programable random-access ability. The snapshot filtering operation does not require any image reconstruction or processing[34–36], which facilitates direct and on-screen observation of a sequence of monochromatic images. Typically, the wavelength switching can be completed in several µs, which is substantially faster than the reported schemes based on OPO tuning[33], grating rotation[37], or mechanical scanning[41]. Moreover, a unique feature of the inertia-free AOTF is the spectral hopping flexibility to fast record images at discrete frequencies over a broadband spectral region, which would be beneficial for high-speed material identification and classification[23]. It is worth noting that the filtering operation is conducted after the frequency upconversion, which allows access to high-performance and cost-effective AOTF in the visible or near-infrared bands. The presented strategy is particularly attractive for high-speed and high-definition hyperspectral imaging at longer infrared wavelengths[45,46], where sensitive imagers and fast filters are typically hard to access.

Moreover, the intrinsic coupling between the space and spectrum during the nonlinear upconversion provides a possibility for leveraging the multiplexing capability, with an aim to further improve the acquisition speed or eventually to implement full-snapshot hyperspectral imaging. In this case, the AOTF is operated at the parallel filtering modality, which allows us to simultaneously record monochromatic images at multiple spectral channels within a single exposure of the camera. The total acquisition time for a complete spectral data cube can thus be significantly shortened to a divided fraction by the number of the multiplexed channels. The underlying challenge for the snapshot operation lies in the extraction of the spectral imaging information[47,48]. As a proof-of-principle demonstration, an iterative algorithm is used to reconstruct the input dispersed image as a multispectral cube, which allows us to obtain clear monochromatic images without dispersive blurring (see Supplementary Note 6).

We note that the presented spectral imaging modality is ready to include the depth information by tuning the pump pulse delay for the optical gating[42,49], which enables rapid data acquisition in all four dimensions, i.e., a $(x, y, z, \lambda)$ hypercube. The envisioned MIR spectrotomography would pave the way for high-speed and high-throughput characterization of biological and material specimens.

## Methods

### Laser sources

Synchronous MIR and pump laser sources are prepared to facilitate the coincidence-pumping configuration, where the pulsed excitation favors increasing the conversion efficiency due to the intensive peak power and suppressing the background noise via the ultrashort optical gating. The MIR source stems from a supercontinuum fiber laser (Novae, Coverage), which is specified with a spectral coverage of 1.9–3.9 µm at a repetition rate up to 2.35 MHz. After spectral and polarization filtering, the MIR beam is expanded to approximately 4 cm in diameter, which results in a spectral intensity is about 7 µW/nm/cm². A small fraction of the infrared beam is detected by a free-space photodetector to trigger an electric pulse generator (Agilent, 81130A). The output TTL signal is then used to modulate a laser diode (LD-PD Inc., PL-DFB-1064) for generating synchronous pump pulses at 1064 nm. The pulse duration of 200 ps is designated to fully enwrap the MIR signal. Meanwhile, a narrow spectral width of 0.2 nm can be maintained at an average power of 600 mW after two-stage fiber amplifiers. The peak power of the pulse pump is about 1.3 kW.

### Imaging setup

The CPLN crystal has a linearly increasing poling period from 16 to 24 µm along the length of 10 mm, which supports an operation window over 2.4–5 µm. The crystal thickness of 2 mm allows us to obtain a spatial resolution of about 70 µm, as discussed in Supplementary Note 4. The polychromatic upconverted beam is steered into an AOTF specified with an aperture size of $8 \times 8$ mm², a switching time below 3 µs, and a diffraction efficiency of over 85%. The spectral filtering resolution can be enhanced to 0.9 nm based on a longer acousto-optic crystal, albeit with a slower switching speed (see Supplementary Note 3). The deflection angle of the filtered field is about 3°, which favors a high rejection ratio for the background noise in combination with a group of interference filters. The filtered monochromatic image is finally captured by a silicon-based CMOS camera (Photron, Mini AX200). The full frame of $1024 \times 1024$ pixels can operate at a cutting-edge speed up to 6.4 kHz, which is about two orders of magnitude faster for infrared focal plane arrays. Here, a higher frame rate at 10 kHz is used for a ~500 kilopixel frame with an aim to demonstrate the full potential of high-speed spectral imaging. In the experiment, a transmission imaging modality is adopted due to the easier optics alignment for setting up the parametric upconverter in the free space. In comparison to the reflection fashion, such a modality is more susceptible to the scattering effect, especially for thick and non-uniform samples. We note that the reflective modality is also feasible as long as the illumination is properly set, for instance, resorting to an oblique incidence. To suppress the spatial coherence of the laser source, a rotatory diffuser can be used to smooth the speckles.

### Reporting summary

Further information on research design is available in the Nature Portfolio Reporting Summary linked to this article.

## Data availability

The data that support the findings of this study are available from the corresponding author upon request. Source data are provided in this paper.

## Code availability

The codes for the space-multiplexed snapshot spectral imaging are available from the corresponding author upon request.

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

## Acknowledgements

This work was supported by the National Natural Science Foundation of China (62175064, 62235019, 62035005, 12022411); Shanghai Pilot Program for Basic Research (TQ20220104); Natural Science Foundation of Chongqing (CSTB2023NSCQ-JQX0011, CSTB2022NSCQ-MSX0451, CSTB2022NSCQ-JQX0016); Shanghai Municipal Science and Technology Major Project (2019SHZDZX01); Fundamental Research Funds for the Central Universities.

## Author contributions

K.H. and H.Z. conceived the project and designed the experiments. J.F., R.Q., and K.H. built the system, performed experiments, and processed data. M.Y. and Y.L. built fiber laser sources. E W. analyzed the imaging data. J.F. and K.H. wrote the manuscript draft. All authors were involved in discussions and contributed to the paper editing.

## Competing interests

The authors declare no competing interests.
