## [Peer Review File · Nature Communications]

Wide-field mid-infrared hyperspectral imaging beyond video rateREVIEWER COMMENTS

Reviewer #1 (Remarks to the Author):

The authors present results combining SFG up-conversion technique to detect MIR radiation with visible AOTF. The imaging requires minor post-processing to correct to achromatism of the imaging system (referred as spatial scaling by authors) associated with broad spectral range, but hats off, this is very elegant and straightforward engineering solution that clearly enabled high speed hyperspectral imaging as demonstrated by the data and video. Besides spectral multiplexing in time domain, authors briefly suggested and show spatial multiplexing, which requires advanced image reconstruction and post-processing. My review is brief, as I think the offered solution is of definite interest to MIR imaging community and it is supported by vivid and clear demonstrations. The work has my full support for publication in Nature Communication.

Two minor and related concerns that I would suggest authors to adapt/address:

1. This work relies on very high brightness sources, i.e. sub-W pump and similar power probe. Besides source availability, high brightness MIR source may render the approach very limited in applications as in majority of biochemical application, the main goal of MIR imaging, it will result in sample/tissue damage.
2. Authors did very respectful and brief up to date review of current state of art including similar conversion approaches as SFG, alternative approaches as NTA and Photothermal and conventional approaches as FTIR. However, all previous approaches utilize significantly lower radiation fluxes, three orders of magnitude lower $\mu\text{W}/\text{nm}$ vs mW/nm used by authors. Moreover, the camera used is significantly more advanced (and way more expensive) to any visible camera's used in any alternative approaches mentioned, i.e. repetition rate and sensitivities. These are crucial parameters as all previous works were aimed at applicability, hence not only to demonstrate high imaging speed, but its combination with conventionally available MIR light fluxes (sub $\mu\text{W}/\text{nm}$) and standard cameras (even sub-research grade). Many approaches will demonstrate comparable numbers if similar source and detector arrays will be used. Thus, making direct comparison as in Table S1 is not fully fair. As for the argument of scanning/moving element, it does not hold here as these were not limiting factors for video-rate imaging in any alternative

approaches. For example, photothermal imaging can produce spectral sweep by scanning QCL $\sim 30,000$ cm⁻¹/s and NTA produce spectral sweep at 12,000 cm⁻¹/s. The similar argument goes to resolution.

Reviewer #2 (Remarks to the Author):

The paper presents a study on wide-field Mid-Infrared (MIR) hyperspectral imaging, showcasing an impressive achievement of a very high frame rate. While the results are noteworthy, the primary focus of the work lies in technical enhancements. Notably, the new additions, the MIR SC laser and AOTF, are commercially available, which may limit the novelty from a technical perspective.

The most substantial improvement appears to be in the initial stage of data acquisition. To provide a clearer context for readers, it is advisable for the authors to include a table comparing key parameters with previous works.

In conclusion, I believe that the level of novelty demonstrated in this work may not meet the criteria for publication in Nature Communications."

Reviewer #3 (Remarks to the Author):

Summary

The manuscript presents a wide-field mid-IR hyperspectral imaging system that achieves a high image acquisition rate due to the use of upconversion detection and an acousto-optic tunable filter. Considering the recent publication by the authors (reference 28), and similar publications in the field, it appears that the main novelty of this paper is the use of a commercial AOTF for wavelength selection. In reference 28, the authors have already demonstrated many of the same features presented in this paper, including 15 kHz monochromatic image acquisition with a 1024x1024 pixel camera, "three-dimensional imaging" based on the pump pulse offset, and wide-field imaging. Another difference is the use of a supercontinuum laser as the mid-IR light source, which considering the noise of such sources is not clearly justified. One interesting concept that is explored here is the

spatial dispersion or “blurring” of wavelengths, which the authors suggest could be used as a way to identify wavelengths from a single composite image. This is a fascinating idea, but unfortunately the methods and results are not convincing, and therefore this part becomes disjoint from the rest of the paper. I have added additional comments below, but my initial verdict is that the manuscript requires major revisions before it can be considered for publication in Nature Communications.

Abstract:

- I would suggest using the term “broad” or “wide” rather than “massive” to describe the wavelength extent of spectral bands.
- It is not clear from the abstract what the “intrinsic space-spectrum coupling” is. I would suggest to revise the sentence to clarify that it is related to the angular dependence of phase-matching in the upconversion.

Page 1

- None of the references 8, 12, and 13 cite characterization of gaseous combustion. Please add a relevant reference.

Page 2:

- I believe the authors meant to say that real-time hyperspectral imaging (i.e. multiple wavenumbers) is NOT possible, but the sentence conveys the opposite meaning, which is that real-time imaging is ONLY permitted in the multi-spectral modality.
- Please define what is considered “video-rate” and “high definition”. Reference 34 achieved 100 Hz frame rate with 640x480 pixels. Is this not considered video-rate and high-definition?
- Regarding description of ref. 30: A 2.5 ms acquisition time is significantly beyond video rate, and the authors mention that if they had a faster camera they could go to 1ms (1 kHz frame rate) based on the need to attenuate the signal. The authors should also refer to this as “beyond video rate”.
- At the end of the this state-of-the-art section, the authors should cite the use of AOTFs for mid-IR hyperspectral imaging performed by M. Farries et al (Proc. of SPIE Vol. 10060, 100600Y-1, 2017) and I. Lindsay et al. (Proc. of SPIE Vol. 9703 970304-1, 2016). In the former, a 100 wavelength x 300k pixel cube was acquired in 2 seconds. The idea with using AOTFs for agile wavelength selection was also presented here.
- The authors should describe in greater detail why wavelength tuning using an AOTF is faster than wavelength tuning of an EC-QCL. Please include cited numbers on scanning

values.

Page 3

- Missing “not” in: “... simultaneous broadband capability has not yet been revealed in...”
- The concept of “spectral transduction” is not clear. Please revise and elaborate on what is meant by this.
- The definition of the abbreviation AOTF should be made at the first use in the introduction.

Page 4

- OPA/OPO tunable sources are not limited to kHz. MHz OPA/OPO sources are commercially available. But since OPA/OPO is slowly tunable, they are not efficient for broadband imaging.
- It is not explained or justified why a pulsed 1064 nm pump is used instead of the simpler CW approach. Please discuss this choice of pumping scheme, its pros and cons.
- From the transmission modality it is clear that the sample has to be very thin, smooth, and uniform to avoid scattering. Otherwise the image will be degraded. Why use a transmission modality instead of the more common reflection modality? How do you avoid speckle and other coherence artefacts?

Page 5

- Fig. 3: The images do not appear to have the usual Gaussian intensity profile associated with upconversion of fiber lasers. Was the beam diffused or otherwise shaped before illuminating the sample or is the beam just much larger than the resolution target?

Page 6

- Please specify what types of plastic films are used in the sample of Fig. 4A.
- Regarding ethyl experiment: This shows that you can image specific chemicals from their absorption, similar to the polymer films, but this does not demonstrate a quantitative method for determining the concentration. Please revise this part.

Page 7

- Fig. 5: This figure is unnecessarily large. The concept in 5A could be shown with just four or five images in a row, as in Fig. 5B.

Page 8

- Regarding the cuvette absorption: The cuvette is clearly transparent at all wavelengths. I suppose you are referring to the edge of the cuvette, which may be made from some polymer that absorbs the mid-infrared light. This should be made clear.

- Since the shadowgraph is monochromatic, it does not reveal any spectral information in the dynamics. Please revise.

Page 9

- The entire section about the space-multiplexed snapshot is very unclear. The authors suggest using the radial dispersion of the image to retrieve the wavelength information, but this must require some predetermined knowledge about the morphology of the sample. Also, since all channels are potentially overlapping there is an inherent ambiguity in the images, especially in more complex samples. I understand the concept that the authors are presenting, but it is not clear from Supplementary section 6 how the authors arrived at images S8.B-G from S8.A. The authors are simply stating that these are reconstructed, but provide no details on how this was done. I understand that the distance from the central "E" is used, but how is the image of a single wavelength channel extracted from the composite image, which is a superposition of all overlapping channels? At this point the image is a matrix of pixel values with no information about the pixel value at different wavelengths, so extracting a single wavelength is not trivial, especially when considering differences in absorption and scattering. It seems to me that this is a very difficult thing to do, and the explanation given by the authors is overly simplified. I would suggest to either leave this part out of the paper, or spend a much greater part of the paper on this to make sure that the methods and results are convincing.

- Almost the entire "discussion" section repeats points from the introduction, or summarizes the methods and results of the paper. There are no new ideas, questions, or issues being discussed. I believe the discussion is missing a paragraph about:

o A) What is the limiting factor or bottle neck in the imaging speed presented in this work, and how to fix it.

o B) The upconversion enables the use of visible light AOTF and silicon cameras, but what is the conversion efficiency? How can it be improved?

o C) The limiting factor in SNR is most likely the supercontinuum laser. What can be done to improve the noise of the system?

o D) The widefield images display a series of rings, which are not part of the imaged sample. Describe where these come from and how can these be removed to obtain higher quality images?

o E) What is the limit of the system in terms of sampling. Obviously the system is sensitive to

scattering and sample thickness, but this should be discussed in greater detail here and compared to other state-of-the-art systems based on e.g. reflection modality.

- The authors claim that the upconverter allows for an extended field of view in one shot. However, it is not clear how the upconversion has anything to do with the field of view, or why parameter scanning and post-processing would help. If anything, the CLNP is a limiting aperture of your system. Please elaborate.
- It is not clear what “globally captured” images refer to.
- Parametric fluorescence is not an accurate term for the nonlinear processes in OPA/OPO sources. I would suggest parametric generation or simply OPO/OPA.
- The concept of using an AOTF is highlighted several times as a key part of this work, but this is not novel in the context of hyperspectral imaging, and there are no novel methods applied to how the AOTF operates.

Page 10

- Biological samples is most likely not possible to image because of high absorption in mid-IR, and even if the sample is dried, it would have to be a very thin slice to not suffer from scattering and speckle. The authors should provide more specific examples of where this instrument has a potential for outperforming existing technologies.
- The first 12 lines of the “laser sources” methods are redundant. Same for the last 3 lines. It does not describe the methods used or the equipment.
- A 200 ps pulse with 0.2nm bandwidth has a time-bandwidth product of about 10, indicating that it is highly chirped. The authors should include in the discussion why a pulsed pump was chosen, and whether the pulse chirp influence the efficiency of the process. Please include the peak power of this pulsed pump source.
- Please add details about sampling methods to the “Imaging setup” section.

References

- The authors should consider citing the following paper for video-rate photothermal imaging: J. Yin et al. ,”Video-rate mid-infrared photothermal imaging by single-pulse photothermal detection per pixel”, Sci. Adv.9, eadg8814(2023).

Manuscript NCOMMS-23-47069-T
“Wide-field mid-infrared hyperspectral imaging beyond video rate”

Reply to the Reviewers

We would like to thank the three reviewers for the careful reading of the manuscript and their valuable reports. We give below a detailed response to the reviewers' comments. Excerpts from the original reports are given in blue. Changes in the revised manuscript are indicated in green.

Reviewer #1

The authors present results combining SFG up-conversion technique to detect MIR radiation with visible AOTF. The imaging requires minor post-processing to correct to achromatism of the imaging system (referred as spatial scaling by authors) associated with broad spectral range, but hats off, this is very elegant and straightforward engineering solution that clearly enabled high speed hyperspectral imaging as demonstrated by the data and video. Besides spectral multiplexing in time domain, authors briefly suggested and show spatial multiplexing, which requires advanced image reconstruction and post-processing. My review is brief, as I think the offered solution is of definite interest to MIR imaging community and it is supported by vivid and clear demonstrations. The work has my full support for publication in Nature Communication.

Two minor and related concerns that I would suggest authors to adapt/address:

We thank the reviewer for his/her positive evaluation on our work. We have carefully revised our manuscript according to the given valuable comments.

1. This work relies on very high brightness sources, i.e. sub-W pump and similar power probe. Besides source availability, high brightness MIR source may render the approach very limited in applications as in majority of biochemical application, the main goal of MIR imaging, it will result in sample/tissue damage.

In our experiment, a commercially available supercontinuum source is used for high-brightness MIR illumination, which aims to provide a sufficient photon flux per acquisition frame as typically required in high-speed imaging. In comparison to direct imaging based on infrared imagers, the implemented upconversion imaging system already enables us to substantially reduce the illumination power due to the enhanced detection sensitivity based on silicon cameras. We would like to note that the probe light is spread into a wide-field viewing area with a diameter about 4 cm, which results in a spectral intensity about $7 \mu\text{W}/\text{nm}/\text{cm}^2$. The modest illumination strength is comparable to the values in other spectral imaging systems, as detailed in the updated version of Supplementary Table S1.

We agree with the reviewer that it is important to control the illumination intensity in biochemical imaging. One solution is to further increase the conversion efficiency by augmenting the pump power.

Another possible solution can resort to the spatial multiplexing scheme as proposed in our work, where multiple spectral components can be reconstructed from a single-shot acquisition. Hence, the exposure time for each shot can be longer, which allows to further reduce the illumination power.

In Supplementary Note 2 in the revised manuscript, we have added related discussion on this point: *“After the spectral and polarization filtering, the maximum illumination power on the sample is about 113 mW. Then, the MIR beam is expanded to approximately 4 cm in diameter through a beam expander, which results in a spectral intensity is about 7 $\mu\text{W}/\text{nm}/\text{cm}^2$.”* And in Supplementary Note 7, we have added: *“Notably, it is important to control the illumination intensity in biochemical imaging. Here the illumination strength is about 7 $\mu\text{W}/\text{nm}/\text{cm}^2$, which is comparable to the values in other spectral imaging systems. The illumination power could further be reduced by improving the conversion efficiency, or resorting to the spatial multiplexing scheme.”*

2. Authors did very respectful and brief up to date review of current state of art including similar conversion approaches as SFG, alternative approaches as NTA and Photothermal and conventional approaches as FTIR. However, all previous approaches utilize significantly lower radiation fluxes, three orders of magnitude lower $\mu\text{W}/\text{nm}$ vs mW/nm used by authors.

We thank the reviewer for this insightful comment. In our work, the illumination power is adapted to demonstrate high-speed and wide-field spectral imaging. Taking into account the large viewing field and broad spectral coverage, the spectral intensity for the probe is about 7 $\mu\text{W}/\text{nm}/\text{cm}^2$, which is modest among reported values. We would like to note that the frame rate of monochromatic acquisition is typically below kHz level in previous imaging systems, which is orders of magnitude slower than the achieved rate here. Therefore, the radiation flux at each frame in our work is much lower than previously reported values. In Supplementary Table S1, we have revised to include the illumination parameter to make a more comprehensive comparison.

Moreover, the camera used is significantly more advanced (and way more expensive) to any visible camera's used in any alternative approaches mentioned, i.e. repetition rate and sensitivities. These are crucial parameters as all previous works were aimed at applicability, hence not only to demonstrate high imaging speed, but its combination with conventionally available MIR light fluxes (sub $\mu\text{W}/\text{nm}$) and standard cameras (even sub-research grade). Many approaches will demonstrate comparable numbers if similar source and detector arrays will be used. Thus, making direct comparison as in Table S1 is not fully fair.

We agree with the reviewer that it is difficult to state which scheme is better than others due to the diversity of application scopes. Our work aims to propose and implement a hyperspectral imaging scheme that is featured with an unprecedented acquisition rate for the data cubes. In our experiment, the cutting-edge light source and optical imager are chosen to demonstrate the full potential in high imaging sensitivity and fast spectral tuning. The presented scheme is compatible for conventional light sources and standard cameras for the sake of improving its applicability.

In previous reports, the spectral imaging speed is typically impeded from the deficiencies in fast wide-

field detection and/or rapid spectral filtering. For instance, Ref. [R1] relies on crystal orientation to achieve wide-field operation, Ref. [R2] relies on grating scanning for spectral selection. Ref. [R3] offers an attractive scheme to implement wide-field MIR spectral imaging based on a high-definition InGaAs camera. However, the imaging sensitivity is inherently limited by the third-order nonlinearity of the detector material. Additionally, the refreshing rate during the continuous acquisition of data cubes is limited by the involved mechanical scanning for the spectral tuning operation. Therefore, even equipped with a high-brightness illumination and a high-speed imager, it is not straightforward for these approaches to achieve comparable spectral imaging speed as demonstrated in our work.

We agree with the reviewer that the cited values in Table S1 cannot fully represent the ultimate performances of the listed schemes. Instead, the given comparison is used to highlight the achieved advance in spectral imaging speed among currently reported values in various schemes for MIR hyperspectral imaging. For the sake of fairness, we have tried to cover the key imaging parameters in the table. It can be seen that our work favors for a high acquisition speed, but some performances still need to be optimized, such as spectral resolution and spectral coverage. In the revised manuscript, we have added related discussions on the future optimization.

[R1] S. Junaid, S. C. Kumar, M. Mathez, M. Hermes, N. Stone, N. Shepherd, M. Ebrahim-Zadeh, P. Tidemand-Lichtenberg, and C. Pedersen, "Video-rate, mid-infrared hyperspectral upconversion imaging," *Optica* 6, 702-708 (2019).

[R2] Y. Zhao, S. Kusama, Y. Furutani, W.-H. Huang, C.-W. Luo, and T. Fuji, "High-speed scanless entire bandwidth mid-infrared chemical imaging," *Nat. Commun.* 14, 3929 (2023).

[R3] D. Knez, B. W. Toulson, A. Chen, M. H. Ettenberg, H. Nguyen, E. O. Potma, and D. A. Fishman, "Spectral imaging at high definition and high speed in the mid-infrared," *Sci. Adv.* 8, eade4247 (2022).

As for the argument of scanning/moving element, it does not hold here as these were not limiting factors for video-rate imaging in any alternative approaches. For example, photothermal imaging can produce spectral sweep by scanning QCL $\sim 30,000$ cm⁻¹/s and NTA produce spectral sweep at 12,000 cm⁻¹/s. The similar argument goes to resolution.

We agree with the reviewer that the QCL and NTA schemes can facilitate fast spectral scanning and high spectral resolution. However, it is currently challenging for both approaches to realize the video-rate hyperspectral imaging at high definition. Specifically, the spectral imaging speed for QCL scheme is limited by the frame rate of the infrared cameras [R4]. Moreover, it typically requires a mosaicking operation to achieve a higher definition due to the limited number of pixels of the infrared detector array, which may extend the overall collection time to several minutes.

Alternatively, NTA scheme provides an attractive solution to access high-definition and high-speed cameras, yet the refreshing rate for the datacube acquisition is limited by the involved mechanical scanning for the spectral tuning operation. Indeed, the spectral scanning rate is fast during a single sweep, yet the dwell time between sequential sweeps is much longer due to the mechanical inertia. In contrast, the inertia-free AOTF used in our work allows for a fast and continuous spectral scanning.

Notably, the electronic filter also supports a programable random-access ability and a multi-wavelength filtering capability, which could be used to facilitate flexible filtering functionalities in spectral imaging acquisition.

In the Supplementary Note 7, we have added related discussions to clarify these points: *“Alternatively, quantum cascade lasers (QCLs) have been used for MIR spectral imaging due to the high illumination brightness, high spectral resolution, and fast wavelength tunability. ..., the performance is currently limited by the infrared imagers that have long been plagued with high dark noise, low pixel count, and limited frame rate.” and “Another attractive conversion method exploits the intrinsic optical nonlinearity of the detector material itself based on non-degenerate two-photon absorption (TPA). The spectral scanning rate can be fast during a single sweep, yet the dwell time between sequential sweeps is much longer due to the mechanical inertial of the involved translational stage.”*

[R4] K. Yeh, S. Kenkel, J.-N. Liu, R. Bhargava, Fast infrared chemical imaging with a quantum cascade laser. Anal. Chem. 87, 485-493 (2015).

Reviewer #2

The paper presents a study on wide-field Mid-Infrared (MIR) hyperspectral imaging, showcasing an impressive achievement of a very high frame rate. While the results are noteworthy, the primary focus of the work lies in technical enhancements. Notably, the new additions, the MIR SC laser and AOTF, are commercially available, which may limit the novelty from a technical perspective.

We thank the reviewer for recognizing the achieved advance in significantly boosting the frame rate for the MIR hyperspectral imaging. Our work reports on the implementation of a MIR spectral upconversion imaging system at high definition and high speed. The implemented imaging system enables us to address the long-standing quest for realizing video-rate acquisition of MIR hyperspectral data cubes with large spatial formats and broad spectral channels.

Here, the full infrared information in spatial and spectral domains is completely transferred into the visible upconversion replica in a single shot, which contrasts to reported schemes that require parameter scanning and post-processing [e.g. Nat. Photon. 6, 788 (2012) and Optica 6, 702 (2019)]. Notably, our work represents the pioneering demonstration of MIR wide-field hyperspectral imaging based on parametric upconversion architecture, which is beyond the achieved imaging dimensions in our previous works [e.g. Nat. Commun. 13,1077 (2022)].

Another key advance lies in the significant boost for the filtering speed by using an acousto-optic tunable filter (AOTF). The inertia-free wavelength switching operation can be cycled in a non-stop and high-speed way, which is much faster than the spectral filtration in recent instantiations based on

grating rotation [Nat. Commun. 14, 3929 (2023)] or mechanical scanning [Sci. Adv. 8, eade4247 (2022)]. As a result, the presented MIR hyperspectral imager allows us to acquire 100 spectral bands over 2600-4085 cm^{-1} in 10 ms, importantly in a continuous fashion (without the dead time between sequential sweeps). The refreshing rate for the datacube acquisition is about two orders of magnitude faster than previously reported results at comparable spectral channels and spatial formats.

Furthermore, the intrinsic space-spectrum coupling in the upconversion image is leveraged for the first time to realize a space-multiplexed snapshot spectral imaging, which provides a novel solution to further increase the hyperspectral imaging speed, or eventually to implement a full-snapshot hyperspectral imaging with the help of advanced computational algorithms.

Therefore, the record-high performances cannot simply be achieved by technical extension of previous works, which instead rely on the novel imaging concept and ingenious experiment design to simultaneously realize fast wide-field detection and rapid spectral filtering. As pointed by the reviewer, the main elements are commercially available, which should not degrade the underlying novelty of this work. In contrast, the easy access of these devices may facilitate the practical use in subsequent applications, which would open up new possibilities in high-throughput characterization of dynamic processes in chemical, medical, and bio-related fields.

The most substantial improvement appears to be in the initial stage of data acquisition. To provide a clearer context for readers, it is advisable for the authors to include a table comparing key parameters with previous works.

In conclusion, I believe that the level of novelty demonstrated in this work may not meet the criteria for publication in Nature Communications.

We follow the reviewer's suggestion, and present a performance comparison of representative wide-field MIR hyperspectral imaging systems in Supplementary Table 1. It can be seen that the frame rate of hyperspectral imaging is typically limited to the Hz level in previous demonstrations, regardless of platforms or techniques. In our work, the achieved 100-Hz frame rate of spectral imaging has significantly surpassed the reported benchmarks, which allows us to address the long-standing quest to realize video-rate acquisition of MIR hyperspectral data cubes with large spatial formats and broad spectral channels. The achieved state-of-the-art spectral imaging performances with fast wide-field detection (\sim tens of μs) and rapid spectral filtering ($\sim 3 \mu\text{s}$) not only advance the frontier of high-dimensional MIR imaging, but also stimulate immediate applications to investigate non-repetitive and highly dynamic processes in material and life sciences.

The achieved imaging performance is way beyond the reach of any reported instantiations in various protocols, which hence represents a significant and firm breakthrough for the MIR hyperspectral imager. We believe that the original implementation and the achieved advance in this work fits well the standards of Nature Communications. We thank again the referee for having given us the opportunity to clarify few points of our paper and to elaborate more explicitly the novelty of our work. We hope with these changes he/she might consider our paper for publication.

Reviewer #3

Summary

The manuscript presents a wide-field mid-IR hyperspectral imaging system that achieves a high image acquisition rate due to the use of upconversion detection and an acousto-optic tunable filter. Considering the recent publication by the authors (reference 28), and similar publications in the field, it appears that the main novelty of this paper is the use of a commercial AOTF for wavelength selection. In reference 28, the authors have already demonstrated many of the same features presented in this paper, including 15 kHz monochromatic image acquisition with a 1024x1024 pixel camera, “three-dimensional imaging” based on the pump pulse offset, and wide-field imaging. Another difference is the use of a supercontinuum laser as the mid-IR light source, which considering the noise of such sources is not clearly justified. One interesting concept that is explored here is the spatial dispersion or “blurring” of wavelengths, which the authors suggest could be used as a way to identify wavelengths from a single composite image. This is a fascinating idea, but unfortunately the methods and results are not convincing, and therefore this part becomes disjoint from the rest of the paper. I have added additional comments below, but my initial verdict is that the manuscript requires major revisions before it can be considered for publication in *Nature Communications*.

We thank the reviewer for his/her careful reading of the manuscript and giving us these valuable comments. In comparison to our previous demonstration in reference 28, the main novelty of this work lies in further elevating the imaging dimension based on the parametric upconversion architecture, and demonstrating a wide-field mid-infrared (MIR) hyperspectral imaging at high definition and high speed over a broad spectral coverage. The involved wide-field operation has only been manifested in high-speed MIR imaging within a narrow spectral band, yet the potential of simultaneous broadband capability has not been revealed in the realm of spectral imaging. Here, the full infrared information in spatial and spectral domains is completely transferred into the upconversion replica in a single shot, which contrasts to reported schemes that require parameter scanning of nonlinear crystals [*e.g.* *Nat. Photon.* 6, 788 (2012) and *Optica* 6, 702 (2019)].

Indeed, the AOTF is the key to realizing the fast spectral filtering in our experiment. It is worth noting that the filtering operation is conducted after the frequency upconversion, which allows to access high-performance and cost-effective AOTF in the visible or near-infrared bands. The presented approach is particularly attractive for high-performance hyperspectral imaging at longer infrared wavelengths or terahertz frequencies, where sensitive imagers and fast filters are typically hard to access. The synergic combination of an infrared upconverter and a short-wavelength AOTF provides a novel way to implement a high-speed MIR spectral imaging based on a high-definition silicon-based camera.

A high-brightness supercontinuum source is used for MIR illumination, which aims to provide sufficient photon fluxes for each frame in the high-speed imaging operation. In order to realize a high-efficiency nonlinear upconversion, a synchronous pulsed pump is engineered to provide an intensive peak power. In parallel, a high-fidelity spectral transduction is required to preserve the infrared absorption profile. The pump bandwidth would result in a resolution degradation due to the spectral

convolution deformation. Hence, we have devised a narrow-band pump for improving the spectral correspondence. The involved spectro-temporal engineering of the synchronous pulses is essential for the implementation of hyperspectral upconversion imaging system.

As recognized by the reviewer, a novel and interesting concept of spatial multiplexing is proposed and preliminarily implemented to realize a snapshot multi-spectral imaging. The wavelength-dependent spatial scaling stems from the phase-matching condition during the nonlinear parametric process, which renders a unique feature for the upconversion imaging scheme. This idea is naturally explored due to the availability of broadband light source and parallel filtering capability in our experiment. We agree with the reviewer that this part is not the major focus of this manuscript. This is why most details are given in the supplementary information to make the main text “tight”. We revised the Supplementary Note 6 to further elaborate the related reconstruction processes and experiment results.

In the following, we present the reply and related changes in the revised manuscript.

Abstract:

- I would suggest using the term “broad” or “wide” rather than “massive” to describe the wavelength extent of spectral bands.

We follow the reviewer’s suggestion to replace the term for better accuracy in the revised manuscript.

- It is not clear from the abstract what the “intrinsic space-spectrum coupling” is. I would suggest to revise the sentence to clarify that it is related to the angular dependence of phase-matching in the upconversion.

The sentence has been revised for better clarity as: *“Moreover, the angular dependence of phase matching in the image upconversion is leveraged to realize a snapshot operation with spatial multiplexing for multiple spectral channels, ...”*

Page 1

- None of the references 8, 12, and 13 cite characterization of gaseous combustion. Please add a relevant reference.

A relevant reference [R1] has been added in the revised manuscript.

[R1] M. J. Baier, A. J. McDonald, K. A. Clements, C. S. Goldenstein, and S. F. Son, “High-speed multi-spectral imaging of the hypergolic ignition of ammonia borane,” Proc. Combust. Inst. 38, 4433-4440 (2020).

Page 2:

- I believe the authors meant to say that real-time hyperspectral imaging (i.e. multiple wavenumbers) is NOT possible, but the sentence conveys the opposite meaning, which is that real-time imaging is ONLY permitted in the multi-spectral modality.

We thank the reviewer for pointing this ambiguous expression, we have revised the sentence to be: *“For instance, the frame rate of QCL-based imaging at a single wavenumber reaches to 50 Hz in 640×480 pixels, which permits a real-time visualization of datacubes only for several spectral bands.”*

- Please define what is considered “video-rate” and “high definition”. Reference 34 achieved 100 Hz frame rate with 640x480 pixels. Is this not considered video-rate and high-definition?

The video rate refers to a refreshing rate of spectral imaging datacubes up to 24 Hz, and high definition refers to megapixel scale. In reference 34, the frame rate of 100 Hz corresponded to a speed for capturing monochromatic images, *i.e.*, the dwell time at each spectral channel is 10 ms. The total acquisition time for a complete datacube reached to several seconds due to the limited speed of the grating scanning, which thus impeded the video-rate visualization of hyperspectral imaging.

We have added the related definitions in the revised manuscript as: *“To date, it is still a long-sought-after goal to realize a video-rate MIR hyperspectral imaging at high definition, i.e., reaching a datacube refreshing rate over 24 Hz for a spatial format in a megapixel scale, which calls for faster acquisition and image processing speed.”*

- Regarding description of ref. 30: A 2.5 ms acquisition time is significantly beyond video rate, and the authors mention that if they had a faster camera they could go to 1ms (1 kHz frame rate) based on the need to attenuate the signal. The authors should also refer to this as “beyond video rate”.

Similarly, the frame rate in ref. 30 refers to the monochromatic imaging modality. The spectral sweeping rate was limited by the tuning speed of the used OPO source. The resulting acquisition time for a set of hyperspectral datacube reached to over 10 s. The “volume” rate is thus far from video rates.

We would like to note that it is imperative to simultaneously realize fast wide-field detection and rapid spectral filtering for obtaining a high “volume” rate. In all previous demonstrations for wide-field MIR hyperspectral imaging, regardless of any platforms or techniques, the “volume” rates are typically limited to the Hz level, which are much slower than the achieved 100 Hz in our work at comparable numbers of spatial pixels and spectral channels.

- At the end of the this state-of-the-art section, the authors should cite the use of AOTFs for mid-IR hyperspectral imaging performed by M. Farries et al (Proc. of SPIE Vol. 10060, 100600Y-1, 2017) and I. Lindsay et al. (Proc. of SPIE Vol. 9703 970304-1, 2016). In the former, a 100 wavelength x 300k pixel cube was acquired in 2 seconds. The idea with using AOTFs for agile wavelength selection was also presented here.

We thank the reviewer for pointing out these related works. The work [R3] has already been cited in our manuscript in the part for explaining the AOTF features. To make a more complete review of previous demonstrations, the two works are now cited in the revised manuscript as: *“Such a bottleneck in wide-field acquisition rate is also manifested in the scheme based on a MIR supercontinuum*

source and an acousto-optic tunable filter (AOTF) [R2, R3]. Specifically, the collection of an image cube of 100 wavelengths and 300k pixels takes as long as 2 seconds [R2].”

In these reports, a mid-IR AOTF is required. In contrast, our work allows to access high-performance and cost-effective AOTF in the visible or near-infrared bands. We would like to note that AOTFs beyond 4.5 μm are not commercially accessible. The presented novel strategy here is particularly attractive to implement hyperspectral imaging at longer infrared wavelengths.

[R2] M. Farries, J. Ward, I. Lindsay, J. Nallala, P. Moselund, “Fast hyper-spectral imaging of cytological samples in the mid-infrared wavelength region,” Proc. SPIE 10060, 100600Y (2017).

[R3] I. D. Lindsay *et al.*, “Towards supercontinuum-driven hyperspectral microscopy in the mid-infrared,” Proc. SPIE, 970304 (2016).

- The authors should describe in greater detail why wavelength tuning using an AOTF is faster than wavelength tuning of an EC-QCL. Please include cited numbers on scanning values.

There seems no such statement that the tuning speed for the AOTF is faster than that for the QCL in our manuscript. If any, we are sorry about the confusion. We have well recognized the spectral tuning feature of QCLs, and wrote: *“Recently, an alternative approach based on quantum cascade lasers (QCLs) has emerged to implement MIR spectral imaging at an improved speed due to the agile spectral tuning capability for the light source.”*

In the direct approach based on QCLs, the “volume” rate in the hyperspectral imaging is mainly limited by the frame rate of the used mid-IR cameras, especially for a large spatial format. We would like to note that QCL sources have recently been adopted for the MIR upconversion spectral imaging. Instead of being limited by the spectral tuning rate, the reported frame rate suffers from the relatively long exposure time up to 10 ms [Opt. Express 26, 2203 (2018)].

We have revised the related part to make clearer statements for the schemes based on QCLs.

Page 3

- Missing “not” in: “... simultaneous broadband capability has not yet been revealed in...”

We thank the reviewer, and make the correction.

- The concept of “spectral transduction” is not clear. Please revise and elaborate on what is meant by this.

This term is replaced with *“spectral mapping”* for better clarity, which refers to the wavelength correspondence between the infrared and upconversion spectra.

- The definition of the abbreviation AOTF should be made at the first use in the introduction.

This has been corrected in the revised manuscript.

Page 4

- OPA/OPO tunable sources are not limited to kHz. MHz OPA/OPO sources are commercially available. But since OPA/OPO is slowly tunable, they are not efficient for broadband imaging.

We thank the reviewer, and have revised the related part as: *“The supercontinuum source eliminates the need of tuning operation as typically required for OPO sources in broadband imaging.”*

- It is not explained or justified why a pulsed 1064 nm pump is used instead of the simpler CW approach. Please discuss this choice of pumping scheme, its pros and cons.

In our experiment, the pulsed pump favors to achieve a high peak power up to 1.3 kW, which is beneficial to increase the nonlinear conversion efficiency and improve the energy utilization efficiency. The accompanied complexity is the requirement of pulse synchronization and delay alignment. Indeed, the CW pumping is simpler in terms of experimental settings. But the conversion efficiency is relatively low. Additionally, the noise is more severe due to the high average power.

The choice of pulsed pumping scheme is discussed in Method section: *“Synchronous MIR and pump laser sources are prepared to facilitate the coincidence-pumping configuration, where the pulsed excitation favors to increase the conversion efficiency due to the intensive peak power and to suppress the background noise via the ultrashort optical gating.”*

- From the transmission modality it is clear that the sample has to be very thin, smooth, and uniform to avoid scattering. Otherwise the image will be degraded. Why use a transmission modality instead of the more common reflection modality? How do you avoid speckle and other coherence artefacts?

Indeed, in our proof-of-principle demonstrations, the sample is chosen to be thin film or chemical liquid, which allows us to better characterize the imaging system. As pointed by the reviewer, the test samples are smooth and uniform to reduce the scattering effect. The transmission imaging modality is used due to the easier alignment of optics in the free space. Reflective modality is also feasible as long as the illumination is properly set, for instance resorting to an oblique incidence. To suppress the spatial coherence, one can use a rotatory diffuser to smooth the speckles as demonstrated in Ref. [R4]. Related discussions have been added in Method in the revised manuscript.

[R4] S. Junaid *et al.*, Video-rate, mid-infrared hyperspectral upconversion imaging. *Optica* 6, 702-708 (2019).

Page 5

- Fig. 3: The images do not appear to have the usual Gaussian intensity profile associated with upconversion of fiber lasers. Was the beam diffused or otherwise shaped before illuminating the

sample or is the beam just much larger than the resolution target?

Indeed, the output MIR beam is enlarged by a beam expander in order to cover the resolution target. This information is now given in Supplementary Note 2: *“Then, the MIR beam is expanded to approximately 4 cm in diameter through a beam expander, ...”*

Page 6

- Please specify what types of plastic films are used in the sample of Fig. 4A.

The information is added in the revised manuscript as: *“Film 1 and film 2 are taken from two kinds of adhesive tapes, which are made of biaxially oriented polypropylene and cellulose acetate, respectively.”*

- Regarding ethyl experiment: This shows that you can image specific chemicals from their absorption, similar to the polymer films, but this does not demonstrate a quantitative method for determining the concentration. Please revise this part.

This experiment is conducted to demonstrate the capability for quantifying the concentration of ethyl alcohol (C₂H₅OH). As illustrated in the Fig. 4(F), higher concentration leads to a high absorption. Therefore, the measured absorbance can be used to quantitatively identify the concentration at various depths, as shown in Fig. 4(G).

In the revised manuscript, we elaborate this part as: *“Figure 4(F) shows the absorbance profiles for four representative points at various depths. The measured absorbance at 3320 cm⁻¹ can be used to quantitatively identify the vertical concentration distribution, as shown in Fig. 4(G).”*

Page 7

- Fig. 5: This figure is unnecessarily large. The concept in 5A could be shown with just four or five images in a row, as in Fig. 5B.

We thank the reviewer's suggestion, and redraw the figure for the sake of conciseness, as shown in Figure R1.

Figure R1: Revised version of Figure 5 in the manuscript.

Page 8

- Regarding the cuvette absorption: The cuvette is clearly transparent at all wavelengths. I suppose you are referring to the edge of the cuvette, which may be made from some polymer that absorbs the mid-infrared light. This should be made clear.

The related sentence is clarified as: “..., *the chemical absorption of the cuvette edge is also manifested in the sequence of monochromatic images, which indicates similar absorption bands for hydrocarbons.*”

- Since the shadowgraph is monochromatic, it does not reveal any spectral information in the dynamics. Please revise.

We thank the reviewer, and revised the sentence as: “*The MIR shadowgraph is useful to reveal temporal and spatial information in flow dynamics at a chosen spectral band.*”

Page 9

- The entire section about the space-multiplexed snapshot is very unclear. The authors suggest using the radial dispersion of the image to retrieve the wavelength information, but this must

require some predetermined knowledge about the morphology of the sample. Also, since all channels are potentially overlapping there is an inherent ambiguity in the images, especially in more complex samples. I understand the concept that the authors are presenting, but it is not clear from Supplementary section 6 how the authors arrived at images S8.B-G from S8.A. The authors are simply stating that these are reconstructed, but provide no details on how this was done. I understand that the distance from the central "E" is used, but how is the image of a single wavelength channel extracted from the composite image, which is a superposition of all overlapping channels? At this point the image is a matrix of pixel values with no information about the pixel value at different wavelengths, so extracting a single wavelength is not trivial, especially when considering differences in absorption and scattering. It seems to me that this is a very difficult thing to do, and the explanation given by the authors is overly simplified. I would suggest to either leave this part out of the paper, or spend a much greater part of the paper on this to make sure that the methods and results are convincing.

As the reviewer pointed out, it is generally a non-trivial task to extract contributions at different wavelengths because each pixel in the recorded snapshot contains the information of many spectral bands. In the presented proof-of-principle demonstration, a very simple scenario is considered to illustrate the basic idea of MIR space-multiplexed spectral imaging. Here, we assume that the sample is homogenous and is covered by an engraved mask with binary transmissions. In this case, the spectral and morphological profile can be extracted by using an iterative algorithm. Additionally, the wavelengths are chosen to be well separated, which leads to a more pronounced blurring effect. The larger radial shift for various spectral components favors the image reconstruction.

In general, the spectral and spatial information is coupled together, and the extraction of a multi-spectral cube from the input dispersed gray image is usually complicated in such an ill-posed optimization problem. The assistance with more advanced algorithms may improve the reconstruction quality and applicability, for instance, resorting to computation spectral imaging techniques based deep learning [R5, R6]. In this work, we focus on initially presenting the possible concept with preliminary investigations. Proper modification or further development of suitable post-processing algorithms will be the direction in the future. To this regard, we keep this section short to make the manuscript “tight”, and leave the details for the algorithm implementation in the Supplementary note 6.

We follow the reviewer’s suggestion, and have added much more discussions on the reconstruction conditions and processing steps to convincingly support the presented methods and results.

[R5] X. Hua, Y. Wang, S. Wang, X. Zou, Y. Zhou, L. Li, F. Yan, X. Cao, S. Xiao, D. P. Tsai, J. Han, Z. Wang, S. Zhu, Ultra-compact snapshot spectral light-field imaging. Nat. Commun. 13, 2732 (2022).

[R6] L. Huang, R. Luo, X. Liu, X. Hao, Spectral imaging with deep learning. Light Sci. Appl. 11, 61 (2022).

- Almost the entire “discussion” section repeats points from the introduction, or summarize the methods and results of the paper. There are no new ideas, questions, or issues being discussed. I believe the discussion is missing a paragraph about:

o A) What is the limiting factor or bottle neck in the imaging speed presented in this work, and how to fix it.

In the revised manuscript, we have added related discussion: *“To go beyond the achieved spectral imaging rate, a sufficient radiation flux should be acquired within a shorter frame time, which requires to augment the illumination power or increase the conversion efficiency.”*

o B) The upconversion enables the use of visible light AOTF and silicon cameras, but what is the conversion efficiency? How can it be improved?

In the revised manuscript, we have added related discussion: *“Currently, the conversion efficiency is about 0.01%, which can be improved by increasing the pump peak power or using a longer nonlinear crystal.”*

o C) The limiting factor in SNR is most likely the supercontinuum laser. What can be done to improve the noise of the system?

We suppose that the noise mentioned by reviewer refers to the spectral and power stability of laser system. We have measured output spectrum repetitively, which indicates a stable spectral profile. The power stability is measured to be about 0.5% for a sampling time of 1 s. In our experiment, the acquisition of hyperspectral datacubes is completed in 10 ms. Within such a short window, the laser system should exhibit a good spectral and power stability.

In our experiment, the SNR of acquired spectral images is mainly limited by the detected photons per pixel, which becomes more pronounced for a shorter frame exposure time below tens of μs . As described in the revised manuscript, *“To go beyond the achieved spectral imaging rate, a sufficient radiation flux should be acquired within a shorter frame time, which requires to augment the illumination power or increase the conversion efficiency.”*

o D) The widefield images display a series of rings, which are not part of the imaged sample. Describe where these come from and how can these be removed to obtain higher quality images?

We have added related discussions in Supplementary Note 5: *“The displayed rings may stem from the phase-matching condition for the nonlinear upconversion, where a slight variation of conversion efficiency is presented along the radial direction. This effect can be removed by subtracting the signal image with a reference that is acquired without the presence of samples.”*

o E) What is the limit of the system in terms of sampling. Obviously the system is sensitive to scattering and sample thickness, but this should be discussed in greater detail here and compared to other state-of-the-art systems based on e.g. reflection modality.

We thank the reviewer for this insight suggestion. The related details have been elaborated in the “Imaging setup” section: *“In the experiment, a transmission imaging modality is adopted due to the*

easier optics alignment for setting up the parametric upconverter in the free space. In comparison to the reflection fashion, such a modality is more susceptible to the scattering effect, especially for thick and non-uniform samples. We note that the reflective modality is also feasible as long as the illumination is properly set, for instance resorting to an oblique incidence. To suppress the spatial coherence of the laser source, a rotatory diffuser can be used to smooth the speckles.”

- The authors claim that the upconverter allows for an extended field of view in one shot. However, it is not clear how the upconversion has anything to do with the field of view, or why parameter scanning and post-processing would help. If anything, the CLNP is a limiting aperture of your system. Please elaborate.

In conventional upconversion schemes, the acceptance angle for the nonlinear crystal is typically small due to the narrow phase-matching bandwidth. As a result, the wide-field operation usually relies on parameter scanning to enlarge the phase-matching angles, for instance, resorting to temperature variation or angular rotation of the nonlinear crystal [e.g. Nat. Photon. 6, 788 (2012) and Optica 6, 702 (2019)], as well as spatial translation of the object scene [e.g. Opt. Express 23, 34023 (2015)]. In our work, a chirped-poling nonlinear crystal is used to implement the upconversion imaging, which naturally supports a broad phase-matching bandwidth. In this scenario, MIR signals over a large range of incident angles can be phase-matched with different poling periods. Therefore, the presented upconverter allows us to obtain a wide field of view in one shot without needing parameter scanning.

In the manuscript, the related discussion has been given as: *“However, a scanning procedure and subsequent post-processing are often needed to extract single-wavelength components from the polychromatic upconverted images, such as relying on temperature variation [25, 35] or angular rotation [34] of the nonlinear crystal, as well as spatial translation of the object scene [36].”*

- It is not clear what “globally captured” images refer to.

Following the previous reply, the global operation refers to recording a wide-field upconversion image in one shot without changing the system settings. To make it clearer, the related sentence is revised to be: *“The involved nonlinear upconverter allows us to obtain a wide field of view in one shot, which contrasts to previous works that need parameter scanning and post-processing [33–36]. The wide-field upconverted images are captured by using a fast and sensitive silicon camera with a megapixel frame matrix.”*

- Parametric fluorescence is not an accurate term for the nonlinear processes in OPA/OPO sources. I would suggest parametric generation or simply OPO/OPA.

We corrected the term to *“parametric generation”* for better accuracy.

- The concept of using an AOTF is highlighted several times as a key part of this work, but this is not novel in the context of hyperspectral imaging, and there are no novel methods applied to

how the AOTF operates.

We agree with the reviewer that the AOTF has been used to implement hyperspectral imaging. However, the operation window is located at mid-IR range to adapt the illumination source. As described in the manuscript, the novelty here is that *“The filtering operation is conducted after the frequency upconversion, which allows to access high-performance and cost-effective AOTF in the visible or near-infrared bands. Notably, AOTFs beyond 4.5 μm are not commercially accessible. The presented strategy is particularly attractive for high-speed and high-definition hyperspectral imaging at longer infrared wavelengths, where sensitive imagers and fast filters are typically hard to access.”*

Page 10

- Biological samples is most likely not possible to image because of high absorption in mid-IR, and even if the sample is dried, it would have to be a very thin slice to not suffer from scattering and speckle. The authors should provide more specific examples of where this instrument has a potential for outperforming existing technologies.

We have added related discussions in Supplementary Note 7: *“Here the illumination strength is about 7 $\mu\text{W}/\text{nm}/\text{cm}^2$, which is comparable to the values in other spectral imaging systems. The illumination power could further be reduced by improving the conversion efficiency, or resorting to the spatial multiplexing scheme. To suppress the spatial coherence of the laser source, a rotatory diffuser can be used to smooth the speckles [12]. Therefore, the achieved features of high acquisition rate, wide-field operation, and broadband spectral coverage would now open up new possibilities in high-throughput characterization of dynamic processes in chemical, medical, and bio-related fields.”*

- The first 12 lines of the “laser sources” methods are redundant. Same for the last 3 lines. It does not describe the methods used or the equipment.

We thank the reviewer’s suggestion, and remove the redundant lines in this section.

- A 200 ps pulse with 0.2nm bandwidth has a time-bandwidth product of about 10, indicating that it is highly chirped. The authors should include in the discussion why a pulsed pump was chosen, and whether the used chirp influence the efficiency of the process. Please include the peak power of this pulsed pump source.

The spectral bandwidth of the pump is determined by the used laser diode. The chirp should not influence the efficiency of the nonlinear process. It is the peak intensity of the pump pulse that affects the conversion efficiency in the sum-frequency generation in our experiment. In Method section on laser sources, we have added related discussions on the choice of pulsed pump and on the used peak power: *“Synchronous MIR and pump laser sources are prepared to facilitate the coincidence-pumping configuration, where the pulsed excitation favors to increase the conversion efficiency due to the intensive peak power and to suppress the background noise via the ultrashort optical gating.”* and *“The peak power of the pulse pump is about 1.3 kW.”*

- Please add details about sampling methods to the “Imaging setup” section.

We thank the reviewer’s suggestion, and have added related details in the Method section.

References

- The authors should consider citing the following paper for video-rate photothermal imaging: J. Yin et al. ,”Video-rate mid-infrared photothermal imaging by single-pulse photothermal detection per pixel”, Sci. Adv.9, eadg8814(2023).

We thank the reviewer for pointing this recent work, we have cited it in the revised manuscript.

REVIEWER COMMENTS

Reviewer #1 (Remarks to the Author):

The authors have carefully addressed the majority of the concerns raised. However, I respectfully disagree with the final statement appended to Supplementary Information Note 7 regarding spectral scan speed of both QCL and NTA.

To elaborate, in the realm of photothermal imaging, the absence of infrared imagers eliminates the need of mosaicking. In this context. In combination with photothermal effect, QCL plays pivotal role in achieving video-rate speeds. Furthermore, it is crucial to clarify that spectral sweep for NTA is also not limited by mechanical scanning limitations. Quite the contrary, the utilization of simple oscillating delay lines allows for spectral scans exceeding 50 Hz over 15 ps range ($>1000\text{ cm}^{-1}$) with an impressive 10 fs precision. I suggest a correction to the aforementioned statement, emphasizing the real-time capabilities of both QCL in photothermal imaging and NTA.

Reviewer #2 (Remarks to the Author):

I understand that the novelty of this work is the high imaging speed achieved with the MIR SC source and AOTF. However, I do not see the fundamental difference of the concept from ref. 12. In particular, the post-processing of the hyperspectral image is as complex as that of Ref. 12. The demonstrated images are only test targets and liquid mixture, which does not show the potential of measuring biological tissues as the authors claim. There is a great variety of samples in life sciences. If you apply the system to a different sized sample, you have to recalibrate everything. Such a feature is not very appropriate for biological tissues. I still do not see that the novelty of this work meets the criteria for publication in Nature Communications.

Reviewer #3 (Remarks to the Author):

The authors have addressed most of the reviewer critique. However, two issues remain:

(1) It is still not clear from the manuscript how using a broadband SC + upconversion +

visible AOTF should be better than a scanning QCL + upconversion. The authors hint at QCL-based schemes being limited by IR cameras, but if the QCL is similarly upconverted, then this is no longer an issue. In fact, one could leave out the AOTF, making the system throughput higher. This point should be more clearly discussed in the manuscript.

(2) Regarding the space-multiplexed snapshot section. This part is still very disjoint from the rest of the paper, and in the current form would be better suited as part of the discussion. Since the authors have clearly put significant efforts into this part judging by supplementary section 6, I believe this should be better reflected in the manuscript. I would suggest to include Figure S9 in the manuscript, and keep the detailed description of the algorithm etc. in the supplementary section.

If the authors manage to address these, I believe the manuscript would be suitable for publication in Nature Communications.

Manuscript NCOMMS-23-47069A
“Wide-field mid-infrared hyperspectral imaging beyond video rate”
Reply to the Reviewers

We would like to thank the three reviewers for the careful reading of the manuscript and their valuable reports. We give below a detailed response to the reviewers’ comments. Excerpts from the original reports are given in blue. Changes in the revised manuscript are indicated in green.

Reviewer #1

The authors have carefully addressed the majority of the concerns raised. However, I respectfully disagree with the final statement appended to Supplementary Information Note 7 regarding spectral scan speed of both QCL and NTA.

To elaborate, in the realm of photothermal imaging, the absence of infrared imagers eliminates the need of mosaicking. In this context. In combination with photothermal effect, QCL plays pivotal role in achieving video-rate speeds. Furthermore, it is crucial to clarify that spectral sweep for NTA is also not limited by mechanical scanning limitations. Quite the contrary, the utilization of simple oscillating delay lines allows for spectral scans exceeding 50 Hz over 15 ps range (>1000 cm⁻¹) with an impressive 10 fs precision. I suggest a correction to the aforementioned statement, emphasizing the real-time capabilities of both QCL in photothermal imaging and NTA.

We thank the reviewer for the valuable comments, and agree that both approaches hold the potential to realize real-time hyperspectral imaging based on further optimization of experimental setting and acquisition process. Now, the related statements are revised as: *“Notably, the use of QCLs in the MIR photothermal imaging allows to access fast-tuning illumination sources over a wide spectral range. In combination with a fast and sensitive camera, the advent of high-power QCLs at high repetition rate would make it possible to implement the wide-field imaging modality to significantly increase the frame rate.”* and *“It is worth mentioning that the refreshing rate in the NTA-based scheme could be substantially improved by using a voice coil actuator as the delay line, which would facilitate a real-time hyperspectral imaging at high definition.”*

Reviewer #2

I understand that the novelty of this work is the high imaging speed achieved with the MIR SC source and AOTF. However, I do not see the fundamental difference of the concept from ref. 12. In particular, the post-processing of the hyperspectral image is as complex as that of Ref. 12. The demonstrated images are only test targets and liquid mixture, which does not show the

potential of measuring biological tissues as the authors claim. There is a great variety of samples in life sciences. If you apply the system to a different sized sample, you have to recalibrate everything. Such a feature is not very appropriate for biological tissues. I still do not see that the novelty of this work meets the criteria for publication in Nature Communications.

We thank the reviewer for recognizing the achieved advance in the high frame rate for the mid-infrared (MIR) hyperspectral imaging. Indeed, our work and the one in ref. 12 both rely on the upconversion imaging architecture, but the underlying concepts and implementations are fundamentally different. Specifically, the wide-field operation in ref. 12 is realized based on mechanical rotation of the nonlinear crystal. In contrast, we use a chirped-poling crystal to facilitate the wide-field imaging in one shot without any parameter tuning. Consequently, a frame rate over 10 kHz is achieved in our work, which is over one order of magnitude faster than the reported value in ref. 12. More importantly, the illumination source in ref. 12 is based on an OPO, which suffers from a slow tuning speed over a wide spectral range. Instead, the combination of broadband SC source and an agile AOTF in our work allows for a rapid spectral tuning within several μs , which is essential to realize a high refreshing rate for the spectral datacube acquisition.

Our upconversion imaging scheme eliminates the stitching operation or long exposure as required in Ref. 12 for gathering all angular components to obtain a wide-field image. In our work, monochromatic images can be recorded by simply setting the filtering center of the AOTF. Moreover, the hyperspectral datacube can be calibrated based on a simple spatial scaling operation on various spectral components. The calibration procedure only needs to be done once. To interrogate samples with different sizes, the involved lens should be adapted for proper imaging magnification, which may require a recalibration as seen in any other imaging systems. But again, the one-time calibration is simple and fast.

The main breakthrough in our work lies in significantly boosting the frame rate for the MIR hyperspectral imaging. To demonstrate and characterize our high-speed imaging system, highly dynamic scenes based on liquid injection or liquid mixing are configured to illustrate the real-time capturing of hyperspectral datacubes beyond video rate. The achieved high “volume” rate up to 100 Hz is much higher than reported values at the Hz level in previous instantiations, which thus allows us to observe fast-evolving behaviors. The spectral coverage of 2600–4085 cm^{-1} for our imaging system is compatible to that in Ref. 12, which should be able to identify chemical constituents in some biological tissues, for instance, addressing the CH_3 stretching vibration of proteins (2930 cm^{-1}) and the CH_2 stretching vibration of lipids (2850 cm^{-1}). Hence, the achieved features of high acquisition rate, wide-field operation, and broadband spectral coverage open new possibilities for investigating transient processes in material and life sciences.

The main scope of this work is to propose and implement a novel scheme for the high-speed MIR spectral imaging. The superior performances are well characterized by using both static and dynamic samples. We believe that the presented performance will stimulate broad interest and further effort in promoting practical demonstrations. Therefore, the original implementation, the achieved advance, and the subsequent impact render this work fit well the standards of Nature Communications. We hope that the reviewer could reconsider our paper for publication.

Reviewer #3

The authors have addressed most of the reviewer critique. However, two issues remain:

(1) It is still not clear from the manuscript how using a broadband SC + upconversion + visible AOTF should be better than a scanning QCL + upconversion. The authors hint at QCL-based schemes being limited by IR cameras, but if the QCL is similarly upconverted, then this is no longer an issue. In fact, one could leave out the AOTF, making the system throughput higher. This point should be more clearly discussed in the manuscript.

We thank the reviewer for the insightful comment. Indeed, the QCL can be used as a fast-tuning illumination source in the mid-infrared upconversion imaging system. Since the QCL typically delivers relatively long optical pulse at the μs level, a continuous-wave pump is usually adopted to conduct the nonlinear conversion. The resulting conversion efficiency is thus much lower than that for the ultrashort pulse pumping in our case. Consequently, spectral imaging at a high signal-to-noise ratio usually requires a long exposure time, which in turn limits the frame rate. For instance, the exposure time for each spectral band is as long as 10 ms in the QCL-based upconversion imaging [34]. The achieved frame rate is about two orders of magnitude slower than that in our work.

In the revised manuscript, we have elaborated this point as: *“Notably, QCL sources have been adopted for MIR upconversion spectral imaging, but the reported frame rate is limited by a relatively long acquisition time up to 10 ms due to the low conversion efficiency in the continuous-wave pumping scheme [34].”*

[34] S. Junaid, J. Tomko, M. P. Semtsiv, J. Kischkat, W. T. Masselink, C. Pedersen, P. Tidemand-Lichtenberg, “Mid-infrared upconversion based hyperspectral imaging,” *Opt. Express* 26, 2203-2211 (2018).

(2) Regarding the space-multiplexed snapshot section. This part is still very disjoint from the rest of the paper, and in the current form would be better suited as part of the discussion. Since the authors have clearly put significant efforts into this part judging by supplementary section 6, I believe this should be better reflected in the manuscript. I would suggest to include Figure S9 in the manuscript, and keep the detailed description of the algorithm etc. in the supplementary section.

We follow the reviewer’s suggestion to put the relevant section in the discussion part. Moreover, the related discussion on the space-multiplexed snapshot imaging is revised to be more concise in order to make the part more fit to the rest of the paper. Here, we aim to initially present the novel concept along with preliminary investigations. In the presented proof-of-principle demonstration, a very simple scenario is considered to illustrate the basic idea of space-multiplexed spectral imaging. We thus feel that it might be too strong to include Figure S9 in the manuscript. More solid illustrations may await further development of comprehensive imaging model and advanced post-processing algorithms, which are actually our ongoing works. To this regard, we would like to keep this section short to make the manuscript “tight”, and leave all the details on reconstruction processes and experiment results in Supplementary Note 6.

If the authors manage to address these, I believe the manuscript would be suitable for publication in Nature Communications.

We have carefully revised our manuscript according to the reviewer's valuable comments. We hope that our manuscript is now suitable for the acceptance in this journal.